



# Contrasting projection of the ENSO-driven $CO_2$ flux variability in the Equatorial Pacific under high warming scenario

Pradeebane Vaittinada Ayar[1], Jerry Tjiputra[1], Laurent Bopp[2], Jim R. Christian[3], Tatiana Ilyina[4], John P. Krasting[5], Roland Séférian[6], Hiroyuki Tsujino[7], Michio Watanabe[8], and Andrew Yool[9]

[1]NORCE Norwegian Research Centre AS, Bjerknes Centre for Climate Research, Bergen, Norway
[2]LMD-IPSL, Ecole Normale Supérieure / Université PSL, CNRS, Ecole Polytechnique, Sorbonne Université, Paris, PSL University, Paris, France
[3]Canadian Centre for Climate Modelling and Analysis, Victoria, BC, CA
[4]Max Planck Institute for Meteorology, Hamburg, Germany
[5]NOAA/Geophysical Fluid Dynamics Laboratory, Princeton, New Jersey 08540, USA
[6]CNRM, Université de Toulouse, Météo-France, CNRS, Toulouse, France
[7]JMA Meteorological Research Institute, Tsukuba, Ibaraki, Japan
[8]Research Institute for Global Change, Japan Agency for Marine-Earth Science and Technology (JAMSTEC), 3173-25, Showa-machi, Kanazawa-ku, Yokohama, Kanagawa, 236-0001, Japan
[9]National Oceanography Centre, Southampton, UK

**Correspondence:** Pradeebane Vaittinada Ayar, (paya@norceresearch.no;pradeebane@laposte.net)

**Abstract.** The El Niño Southern Oscillation (ENSO) widely modulates the global carbon cycle, in particular, by altering the net uptake of carbon in the tropical ocean. Indeed, over the tropics less carbon is released by oceans during El Niño while it is the opposite for La Niña. Here, the skill of Earth System Models (ESM) from the latest Coupled Model Intercomparison Project (CMIP6) to simulate the observed tropical Pacific $CO_2$ flux variability in response to ENSO is assessed. The temporal

amplitude and spatial extent of $CO_2$ flux anomalies vary considerably among models, while the surface temperature signals of El Niño and La Niña phases are generally well represented. Under historical conditions followed by the high warming Shared Socio-economic Pathway (SSP5-8.5) scenarios, about half the ESMs simulate a reversal in ENSO-$CO_2$ flux relationship. This gradual shift, which occurs as early as the first half of the 21st century, is associated with a high $CO_2$-induced increase in Revelle factor that leads to stronger sensitivity of partial pressure of $CO_2$ ($pCO_2$) to changes in surface temperature between

ENSO phases. At the same time, uptake of anthropogenic $CO_2$ substantially increases upper ocean dissolved inorganic carbon (DIC) concentrations, reducing its vertical gradient in the thermocline, and weakening the ENSO-modulated surface DIC variability. The response of ENSO-$CO_2$ flux relationship to future climate change is sensitive to the contemporary mean state of the carbonate ion concentration in the tropics. Models that simulate shift in ENSO-$CO_2$ flux relationship simulate positive bias in surface carbonate concentration.

## 1 Introduction

Since the beginning of the industrial era, human activities such as fossil fuel combustion, land-use changes and cement production have released huge amounts of greenhouse gases (predominantly $CO_2$) leading to the ongoing planetary scale climate





change. This excess $CO_2$ in the atmosphere is partly absorbed by the ocean and terrestrial biosphere, buffering the rate of warming (Doney et al., 2014; Le Quéré et al., 2016). Over 2010-2019, approximately $3.4 \pm 0.9$ Pg C $yr^{-1}$ and $2.5 \pm 0.6$ Pg C

$yr^{-1}$ are absorbed respectively by the land and ocean, with substantial interannual variability (Friedlingstein et al., 2020). Due to its strong feedback to climate, improved understanding of this variability, governing mechanisms, and how they may evolve in the future are required to constrain future climate change projections.

Due to its vast area, the tropical Pacific is the most important $CO_2$ outgassing region in the world oceans today (Takahashi et al., 2009), representing more than 17% of the global ocean $CO_2$ uptake ($0.44 \pm 0.41$ Pg C per year for 1990-2009 and

18°S-18°N, Ishii et al., 2014) and is projected to be the second region (after the Southern Ocean) with the highest amount of area-integrated anthropogenic carbon uptake in the 21$^{st}$ century under high $CO_2$ scenario (Tjiputra et al., 2010; Roy et al., 2011). In terms of interannual variability, the Equatorial Pacific $CO_2$ flux represents the dominant mode of variability of the global oceanic $CO_2$ flux variations (Wetzel et al., 2005; Resplandy et al., 2015; Landschützer et al., 2016). In this region, the mechanistic driver is associated with the El Niño-Southern Oscillation (ENSO), which has been well established and

thoroughly documented in many previous observational and modeling studies. For instance, Feely et al. (2006) showed strong negative correlation between $CO_2$ fluxes and ENSO over the Equatorial Pacific using observations from 1981 to 2004. Using ocean biogeochemical general circulation models forced with atmospheric reanalysis, similar regional $CO_2$ flux fluctuations in response to ENSO have been simulated (Winguth et al., 1994; Bousquet et al., 2000; Valsala et al., 2014; Wang et al., 2015).

The biogeochemical processes constraining the $CO_2$ fluxes in the Equatorial Pacific are strongly influenced by the ENSO-

induced physical processes. These processes can be formulated as follows: during El Niño events, warmer sea surface temperature reduces the $CO_2$ solubility which increases seawater partial pressure of $CO_2$ ($pCO_2$, Le Borgne et al., 2002; Patra et al., 2005; Ishii et al., 2014). In parallel, during those events, weaker upwelling of nutrient- and dissolved inorganic carbon-rich subsurface water acts to reduce the surface seawater $pCO_2$ (Feely et al., 2006; Long et al., 2013; Wang et al., 2015). The opposite happens during the La Niña phase. Among these competing processes, the ENSO-driven interannual variability of

$CO_2$ flux is presumably dominated by the modulation of dissolved inorganic carbon (DIC) concentration by the upwelling process (McKinley et al., 2004; Li and Xu, 2013; Jin et al., 2017). Therefore, it is the change of thermocline depth and upwelling strength during ENSO phase that mainly govern the tropical Pacific $CO_2$ flux anomalies by constraining on surface DIC concentration (*e.g.,* Doney et al., 2009).

While models simulating ocean only are able to simulate the relationship between $CO_2$ and ENSO (*e.g.* McKinley et al.,

2004; Wetzel et al., 2005; Li and Xu, 2013), this is not always the case for fully coupled Earth system models (ESMs). Indeed, based on ESM simulations from the Coupled Model Intercomparison Project 5 (CMIP5, Taylor et al., 2012), Dong et al. (2017) showed that over the historical period some models underestimate the observed surface DIC variability and consequently the $CO_2$ flux anomalies. They attributed this to a weak relationship between the simulated upwelling variations and the respective ENSO phases. Jin et al. (2019) enlightened that some ESMs poorly simulate the spatial pattern of the tropical

Pacific $CO_2$ fluxes in response to ENSO over the historical period. They attributed this to the weak surface DIC-induced $CO_2$ flux variability during ENSO, e.g. the anomalously low DIC signals associated with ENSO are insufficient to counteract the SST-induced solubility effects.





The main focus of this paper is to determine how the ENSO-induced variability of sea-air $CO_2$ fluxes may be altered in the high-$CO_2$ future in ESM projections. In this study, the capability of the latest ESM collection from CMIP6 (Eyring et al., 2016) in reproducing the observed ENSO-$CO_2$ flux relationship over the contemporary period is first evaluated. Next, we analyze how this relationship evolves over an end member future projection. Given the importance of carbon cycle climate feedback on future projections (e.g., Arora et al., 2020) and the large-scale impact of ENSO on the global climate, such evaluation is timely and necessary. In particular, the aim is to identify and elucidate emerging consistent pattern among the ESMs to better constrain future changes in ENSO-induced variability in the Equatorial Pacific. Studying the future evolution of ENSO-related $CO_2$ flux variations is also crucial since ENSO, the most dominant mode of global climate variability, and its extremes are projected to become more frequent, more intense and more extended in spatial impact (Cai et al., 2015).

The paper is organized as follows. Section 2 introduces the observational and model datasets, the study area, as well as the methods used to analyse the relationship between ENSO and sea-air $CO_2$ fluxes. Results on the contemporary ENSO related spatial patterns and ENSO-$CO_2$ flux relationship reversal and variability drivers are presented in Section 3, while Sections 4 and 5 provide the discussion and summary of this study.

## 2 Data and methodologies

### 2.1 Observational and CMIP6 datasets

The ocean variables analyzed in this study are listed in Table 1. These variables are extracted from different observational and simulation products at monthly temporal resolution. For observational-based fgco₂, the monthly reconstruction values from 1982 to 2015 based on a two-step neural network data interpolation is used (Landschützer et al., 2016). Gridded monthly SST observations are taken from the Japanese 55-year Reanalysis reanalysis data (JRA-55) from 1958 to 2019 (Kobayashi et al., 2015; Harada et al., 2016). The subsurface temperature profiles over the 1985-2014 period are computed from the ORAS5 reanalyses (Zuo et al., 2019). Total alkalinity average estimate including measures between 1972 and 2013 has been retrieved from the GLODAP version 2 data product (Lauvset et al., 2016). Finally, the observed DIC climatology over the 2004-2017 period is extracted from Keppler et al. (2020) dataset. All variables are given at a regular $1° \times 1°$ spatial horizontal resolution.

For the Earth system model simulations, the monthly output fields of surface fgco₂, $pCO_2$, SST, intPP, as well as 3D temperature, DIC and alkalinity concentrations are taken from the Coupled Model Intercomparison Project phase 6 (CMIP6, Eyring et al., 2016) database. At the time of study initiation, sixteen ESMs provide these variables required for the analysis (see Table 2). The simulation variant for each model is chosen according the availability of the variables shown in Table 1. Given the variety of (irregular) grids among the models, the model data sets are spatially regridded into a regular $1° \times 1°$ grid using climate data operators (CDO). The vertical resolutions of 3D temperature and DIC are linearly interpolated at 20 m resolution from the surface down to 1000 m depth.

In this study, analyses are conducted over the same contemporary reference period 1985-2014, the end of the century future period 2071-2100 under the high $CO_2$ Shared Socio-economic Pathway scenario (SSP5-8.5, O'Neill et al., 2016) and the whole 1850-2100 period, combining both historical and SSP5-8.5 concentration-driven experiments. This high warming scenario has





**Table 1.** Ocean variables used in this study. The full name, the abbreviation, standardized CMIP6 name and the unit of each variable is given.

| Variable | abbreviation | standardized name | unit |
|---|---|---|---|
| surface sea-air $CO_2$ fluxes | fgco$_2$ | fgco2 | mol C m$^{-2}$ yr$^{-1}$ |
| surface $CO_2$ seawater partial pressure | $p$CO$_2$ | spco2 | µatm |
| sea surface temperature | SST | tos | °C |
| vertically integrated primary production by phytoplankton | intPP | intpp | mol C m$^{-2}$ yr$^{-1}$ |
| export production at 100m | epc100 | epc100 | mol C m$^{-2}$ yr$^{-1}$ |
| 3D fields of dissolved inorganic carbon concentration | DIC | dissic | µmol C L$^{-1}$ |
| 3D fields of temperature | - | thetao | °C |
| 3D fields of alkalinity | ALK | talk | µmol eq L$^{-1}$ |
| 3D fields of carbonate ion (estimated as ALK-DIC) | $CO_3^{2-}$ | - | µmol C L$^{-1}$ |

been chosen in order to use a clear signal with a high signal to noise ratio. Indeed, using a high emissions end-member scenario gives us the best chance to actually see a change in such strong relationship between ENSO and $CO_2$ fluxes. The model simulation outputs are first evaluated against the observations for the reference period, followed by analysis of future evolution and changes with respect to the reference period.

**Table 2.** List of the 16 CMIP6 models used in this study with the horizontal resolution of the ocean component, variant label, model and data references. Note that most of the models have irregular grids and the resolution quoted in the table are approximate.

| CMIP6 Model Name | Horizontal Ocean Resolution (lon. by lat. in degree) | Variant Label | ESM Reference | Data |
|---|---|---|---|---|
| ACCESS-ESM1-5 | 1°×1° | r1i1p1f1 | Law et al. (2017) | Ziehn et al. (2019) |
| CanESM5-CanOE | 1°×1° | r1i1p2f1 | Swart et al. (2019c) | Swart et al. (2019b) |
| CanESM5 | 1°×1° | r1i1p2f1 | Swart et al. (2019c) | Swart et al. (2019a) |
| CESM2 | 1.125°×0.53° | r10i1p1f1 | Lauritzen et al. (2018) | Danabasoglu (2019a) |
| CESM2-WACCM | 1.125°×0.53° | r1i1p1f1 | Liu et al. (2019) | Danabasoglu (2019b) |
| CNRM-ESM2-1 | .3°-1° | r1i1p1f2 | Séférian et al. (2019) | Seferian (2018) |
| GFDL-CM4 | 0.25°×0.25° | r1i1p1f1 | Held et al. (2019) | Guo et al. (2018) |
| GFDL-ESM4 | 0.5°×0.5° | r1i1p1f1 | Dunne et al. (2020) | Krasting et al. (2018) |
| IPSL-CM6A-LR | .3°-1° | r1i1p1f1 | Boucher et al. (2020) | Boucher et al. (2018) |
| MIROC-ES2L | 1°×1° | r1i1p1f2 | Hajima et al. (2020) | Hajima et al. (2019) |
| MPI-ESM1-2-HR | 0.4°×0.4° | r1i1p1f1 | Müller et al. (2018) | Jungclaus et al. (2019) |
| MPI-ESM1-2-LR | 1.5°×1.5° | r1i1p1f1 | Mauritsen et al. (2019) | Wieners et al. (2019) |
| MRI-ESM2-0 | 1°×(0.3-0.5)° | r1i2p1f1 | Yukimoto et al. (2019a) | Yukimoto et al. (2019b) |
| NorESM2-LM | 1°×1° | r1i1p1f1 | Seland et al. (2020) | Seland et al. (2019) |
| NorESM2-MM | 1°×1° | r1i1p1f1 | Seland et al. (2020) | Bentsen et al. (2019) |
| UKESM1-0-LL | 1°×1° | r1i1p1f2 | Sellar et al. (2019) | Tang et al. (2019) |



## 2.2 Variable anomalies, Niño34 index and thermocline depth computation

The analysis focuses on the correlation between $CO_2$ flux anomalies and Niño34 index. First, the monthly anomalies of sea-air $CO_2$ fluxes at each grid-point are computed by detrending each calendar month separately using a cubic smoothing spline (implemented by the function smooth.spline in R software; R Core Team, 2016) over the period 1850-2100. For instance, the non-linear trend of Januaries at a given grid-point is removed from the respective time-series comprising all January values. The SST and $pCO_2$ anomalies used in the analyses are also computed in the same manner. The degree of freedom of the spline is set to get a good compromise between the smoothness (smoothing parameter above 0.8) and the number of parameters (knots) to estimate the trend over to whole Equatorial Pacific with (Hastie and Tibshirani, 1990, Chap.10). The degree of freedom is set to 5 for SST and fgco$_2$. A degree of freedom of 12 is needed for $pCO_2$ given its steeper increase.

The Niño34 index corresponds to the standardised area-weighted mean SST anomalies over the Niño34 region: 5°S-5°N × 190°-240°E. These anomalies are computed relative to the 1981-2010 climatology. For the CMIP6 model outputs, the SST values are first detrended over the 1850-2100 period using cubic spline. Then, model specific Niño34 index is computed relative to the 1981-2010 climatology. Hereafter, the regimes referred to as El Niño (La Niña) are defined from the respective Niño34 indices (specific for observations and each models). For months with Niño34 index above one standard deviation of each dataset specific Niño34 are categorised into El Niño regime, and vice versa for La Niña regime.

The thermocline is a transition layer where the temperature decreases rapidly with depth from the warm surface mixed layer to the cold deep water layer, where the temperature is relatively uniform. A deeper thermocline (*e.g.,* during El Niño) limits the amount of interior DIC brought to shallower depths by upwelling. This indicator is used in this study to assess the changes in the mechanisms linking ENSO and $CO_2$ fluxes in the present day and in the future projections. The thermocline depth is typically defined as the depth with the maximum vertical temperature gradient (Zhu et al., 2021, and the reference therein). In this paper, the gradient is computed as the vertical difference within each 20m layer (after the vertical interpolation) and the thermocline depth is the average depth of the layer with highest gradient.

## 2.3 Thermal and non-thermal contributions to surface $pCO_2$

In order to differentiate the thermal (th, driven by SST) and non-thermal (nt, driven by other factors, such as DIC, alkalinity and salinity) contributions, the temporal variations of surface ocean $pCO_2$ is decomposed into the two terms following Takahashi et al. (1993; 2002). Seawater $pCO_2$ is thermodynamically dependent on temperature and is computed from the temperature sensitivity of $CO_2$ $\gamma_T$ ($4.23\%°C^{-1}$). This sensitivity has experimentally been determined and is associated with very little error (Takahashi et al., 1993), which is not further considered. The thermal $pCO_2$ component $pCO_2^{th}$ is computed as follows:

$$pCO_2^{th} = < pCO_2 >_{annual} \exp\left(\gamma_T(dSST - < dSST >)\right). \tag{1}$$

In Eq. (1), the annual $pCO_2$ average, $< pCO_2 >_{annual}$ is perturbed with temperature anomalies computed as the difference between the detrended SST, dSST (done with a cubic spline) and the long term mean dSST, <dSST>. The non-thermal component ($pCO_2^{nt}$), which reflects the effect of biophysical processes, is computed by normalizing the $pCO_2$ to <dSST> (Takahashi





et al., 2002):

$$pCO_2^{nt} = pCO_2 \exp\left(\gamma_T(<dSST> - dSST)\right) \tag{2}$$

In Eq. (2), the exponential term removes the SST-associated $pCO_2$ variation. This decomposition is well-known and extensively

used at regional and global scale (*e.g.,* Landschützer et al., 2018; Jiménez-López et al., 2019; Ko et al., 2021).

## 2.4   Biological contribution to surface $pCO_2$

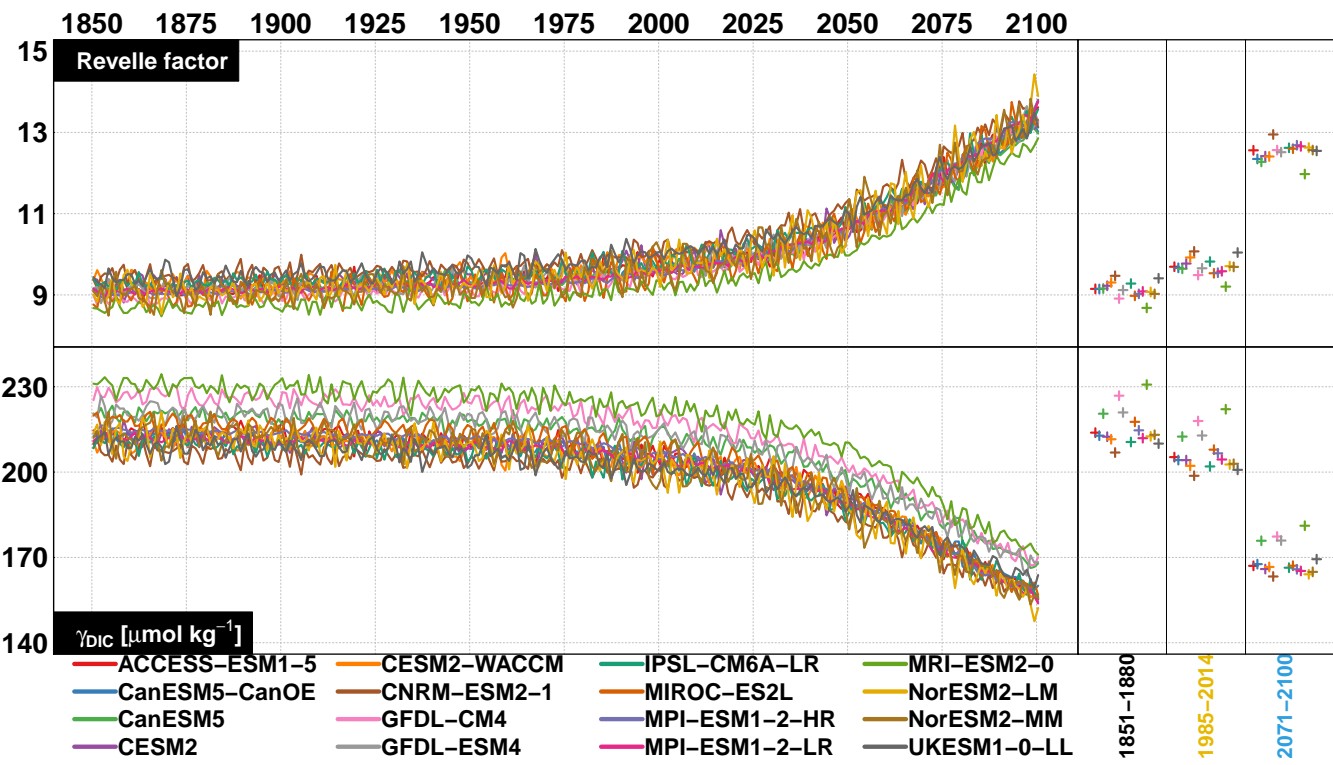

**Figure 1.** Annual Revelle factor and $\gamma_{DIC}$ (in µmol kg$^{-1}$) for CMIP6 models. Average ESMs over the early historical (1851-1880), contemporary (1985-2014 ) and future (2071-2100 ) are given in the right panels.

The buffering capacity of the ocean is a measure of the ability of the ocean to take up carbon and is quantified by the Revelle factor, $R = \frac{\Delta pCO_2}{pCO_2} / \frac{\Delta DIC}{DIC}$ (Revelle and Suess, 1957). The Revelle factor $R$ is the ratio of the relative change of seawater $pCO_2$ (or aqueous $CO_2$ concentration, $CO_{2(aq)}$) to the relative change of dissolved inorganic carbon (DIC $= CO_{2(aq)} + HCO_3^- + CO_3^{2-}$,

Egleston et al., 2010; Hauck and Völker, 2015). The sensitivity of $pCO_2$ to DIC perturbations can be estimated using the buffer factor $\gamma_{DIC}$ that is related to the Revelle factor as $\gamma_{DIC} = \frac{DIC}{R}$ and can be explicitly retrieved from the carbonate system parameters (Egleston et al., 2010). To summarise, the higher the Revelle factor, the lower the buffer capacity (or the buffer factor $\gamma_{DIC}$) of the ocean and its $CO_2$ uptake capacity. The annual evolution of surface Revelle factor and buffer factor $\gamma_{DIC}$ for


CMIP6 models over the 1850-2100 period in the Equatorial Pacific (defined below) are given in Fig. 1. Using this relationship,

the reduction in $pCO_2$ can be quantified as a result of reduction in DIC concentration, e.g., associated with biological carbon

absorption:

$$\Delta pCO_{2\ \text{bio}} = \frac{\Delta \text{DIC}_{\text{bio}}}{\gamma_{\text{DIC}}} pCO_2 \qquad (3)$$

where $\Delta \text{DIC}_{\text{bio}}$ is the mean reduction in surface DIC concentration due to biological production (estimated from the monthly

intPP in [mol C m$^{-2}$ month$^{-1}$] divided by the euphotic layer depth, here assumed to be 100 m). A similar approach has been

used in Hauck and Völker (2015) to determine the impact of biological activity on surface $pCO_2$ in the Southern Ocean. The

$\Delta pCO_{2\ \text{bio}}$ is relevant to evaluate the biological contributions, during El Niño and La Niña, to $pCO_2$ variations. This quantity

non-linearly increases with $\Delta \text{DIC}_{\text{bio}}$, *i.e.*, biological contributions to $pCO_2$ variations increases as the buffering capacity

decreases.

## 2.5 Study Area

For analysis of integrated surface properties, the focus on evaluating the anomalies over the Equatorial Pacific is given within

the 2°S-2°N and 180°-260°E domain (hereafter referred to as Equatorial Pacific or simply EP). EP area is indicated by the

green box in the bottom right SST panel of Figure 2. This region is identified as the common domain where the models and

observation show the largest change in SST between ENSO phases. The same domain is also considered for subsurface analysis

conducted in this study, namely the changes in the vertical DIC, carbonate ion concentration and temperature profiles between

the contemporary and future periods.

## 3 Results

### 3.1 Contemporary (1985-2014) ENSO-related patterns

Figure 2 depicts the tropical Pacific SST and sea-air $CO_2$ fluxes average anomalies for La Niña and El Niño regimes over the

contemporary period from observations and the CMIP6 multi-model mean. The corresponding values for each model are given

in Figs. S1 and S2 of the supplemental material. For surface temperature anomalies, some models clearly simulate too strong

and too broad SST anomalies (Fig. S1) but the CMIP6 multi-model ensemble mean values show a strong resemblance with

the observations, though with slightly too strong anomalies in the central Equatorial Pacific. However, the warm anomalies

observed over the coast of Peru during El Niño is slightly weaker in the model simulations. In these two regions, the inter-

model variability is also large (contour lines in Fig. 2). For the sea-air $CO_2$ flux anomalies, the simulated spatial extent are

less in agreement with the observational estimates. The spatial distribution of $CO_2$ flux anomalies are also different from one

model to another and none of the model simulate a spatial correlation with observation of more than 0.8 according the regime

with even negative correlation (see Fig. S3 of supplemental material). The co-location of spatial distribution of the temperature

and $CO_2$ flux anomalies during the ENSO phase is quite straightforward in the observations while it seems less obvious in

the models. This suggests that some of the observed mechanisms governing the ENSO-related variability of $CO_2$ flux are not



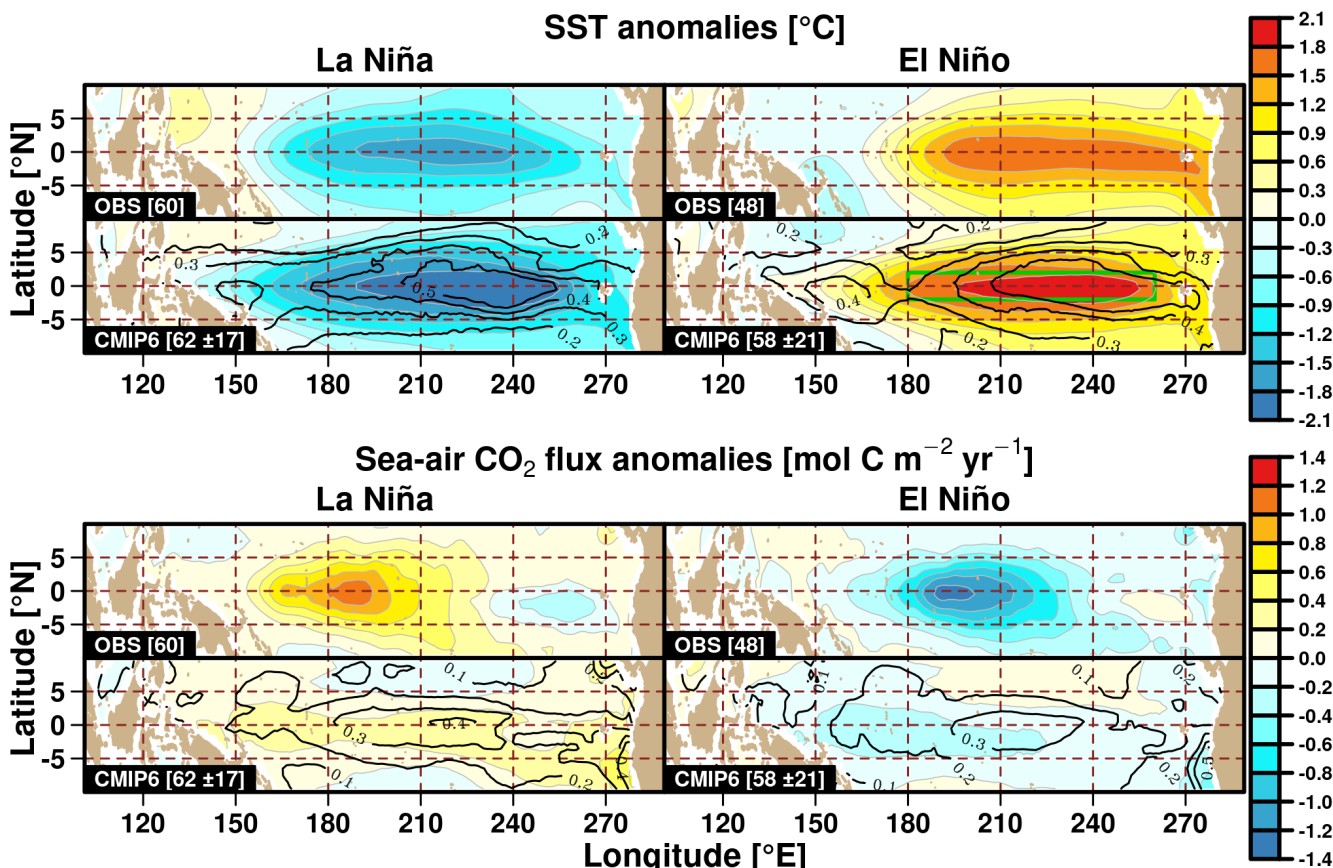

**Figure 2.** Observed and mean CMIP6 SST (in °C, *top*) and sea-air $CO_2$ fluxes (in mol C m$^{-2}$ yr$^{-1}$, *bottom*) average anomalies over the 1985-2014 contemporary period for the La Niña (*left*) and El Niño (*right*) regimes. In squared brackets, the number of months in each regimes are given for the observations and the mean number with one standard deviation for CMIP6 ensemble. Black contours indicate CMIP6 ensemble anomalies one standard deviation. Green box in the lower right SST panel illustrates the EP (Equatorial Pacific) area.

well reproduced by the models. Most models simulate a weaker $CO_2$ flux anomalies compared to the observations, which is consistent with that of CMIP5 model results (Dong et al., 2017). Nevertheless, the multi model mean reproduces the observed outgassing anomaly signals over most of the tropical Pacific during La Niña, and vice versa for El Niño.

Figure 3 shows the zonal average of temperature and DIC vertical sections over the contemporary period and its anomalies during the La Niña and El Niño regimes from the observations and the CMIP6 ensemble mean (for DIC, only the mean values

is shown for observations). During El Niño events, the observations depict a clear warming signal in the eastern part of the tropical Pacific extending throughout the upper ocean with a maximum warming around 70 m depth. Cool anomaly can be seen in the western part of the domain at approximately 150 m depth. The opposite anomaly patterns can be seen during La Niña. The observed and simulated long-term mean temperature patterns are quite similar, while the magnitude of the anomalies are weaker in the CMIP6 multi-model mean. The contemporary DIC average concentration is generally higher in the models than

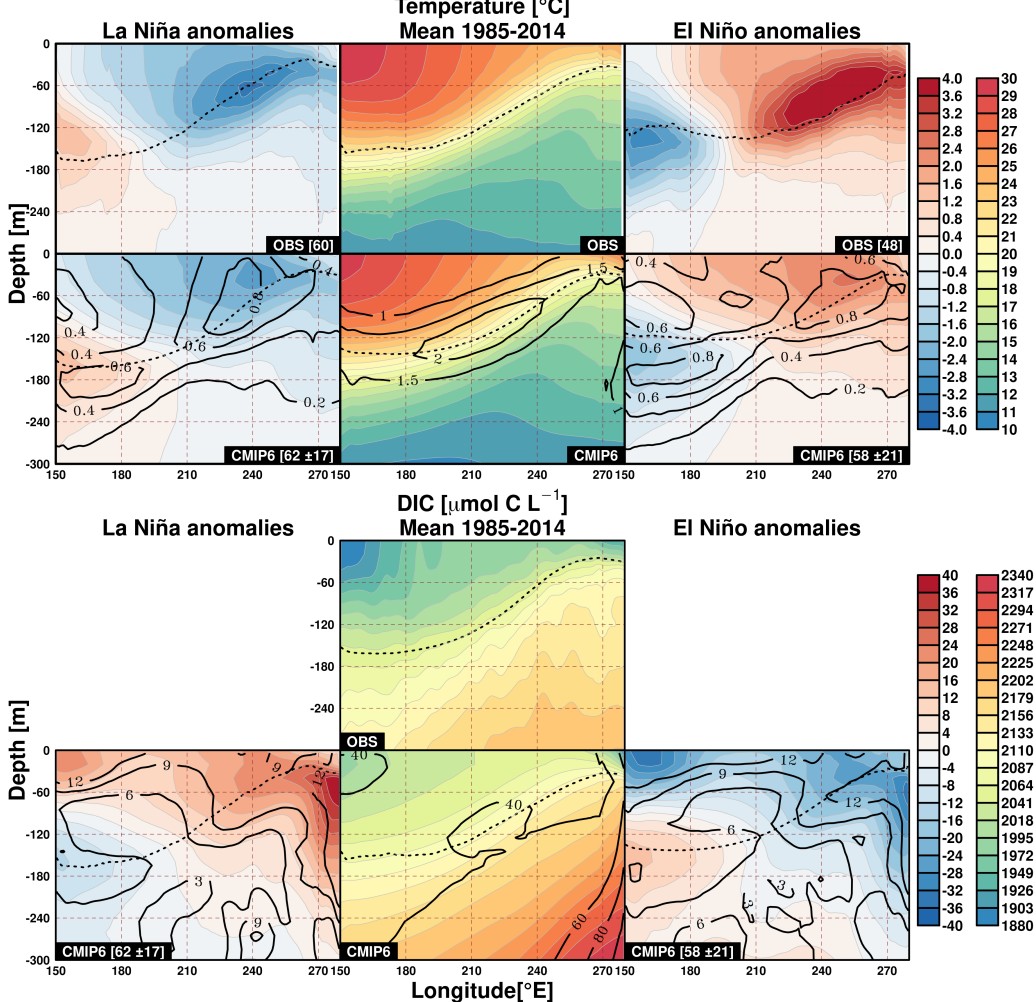

**Figure 3.** Observed and mean CMIP6 vertical section of temperatures (in °C, *top*) and DIC (in µmol C L$^{-1}$, *bottom*) zonal (between 5°N and 5°S) average over the 1985-2014 contemporary period (*middle column*). Average anomalies (differences) relative to contemporary mean are given for La Niña (*left*) and El Niño (*right*) regimes. Dotted lines indicate the average thermocline depth. In square brackets, the number of months in each regimes are given for the observations and the mean number with one s.d. for the CMIP6 ensemble. Black contours indicate CMIP6 ensemble anomalies one standard deviation.

in the observations. Note that the observed average is the result of the climatology over the 2004-2017 period while the average for CMIP6 is computed over 30 years (1985-2014). The subsurface DIC signals of anomalies contrasting La Niña and El Niño regimes are pronounced in the upper layer but also in the east of 240°E down to 300 m, with positive (negative) anomalies during La Niña (El Niño) associated with changes in the upwelling dynamics. This area presents also the largest inter-model variability. Consequently, this DIC anomaly determines the CO$_2$ flux anomaly at the surface. An opposite DIC anomaly signal





is simulated in the western part of the section below 100 m depth. The zonal average of temperature and DIC along the vertical

sections and its anomalies from each individual model are given in Figs. S4 and S5 of the supplemental material.

### 3.2  Transient changes in ENSO-$CO_2$ flux relationship

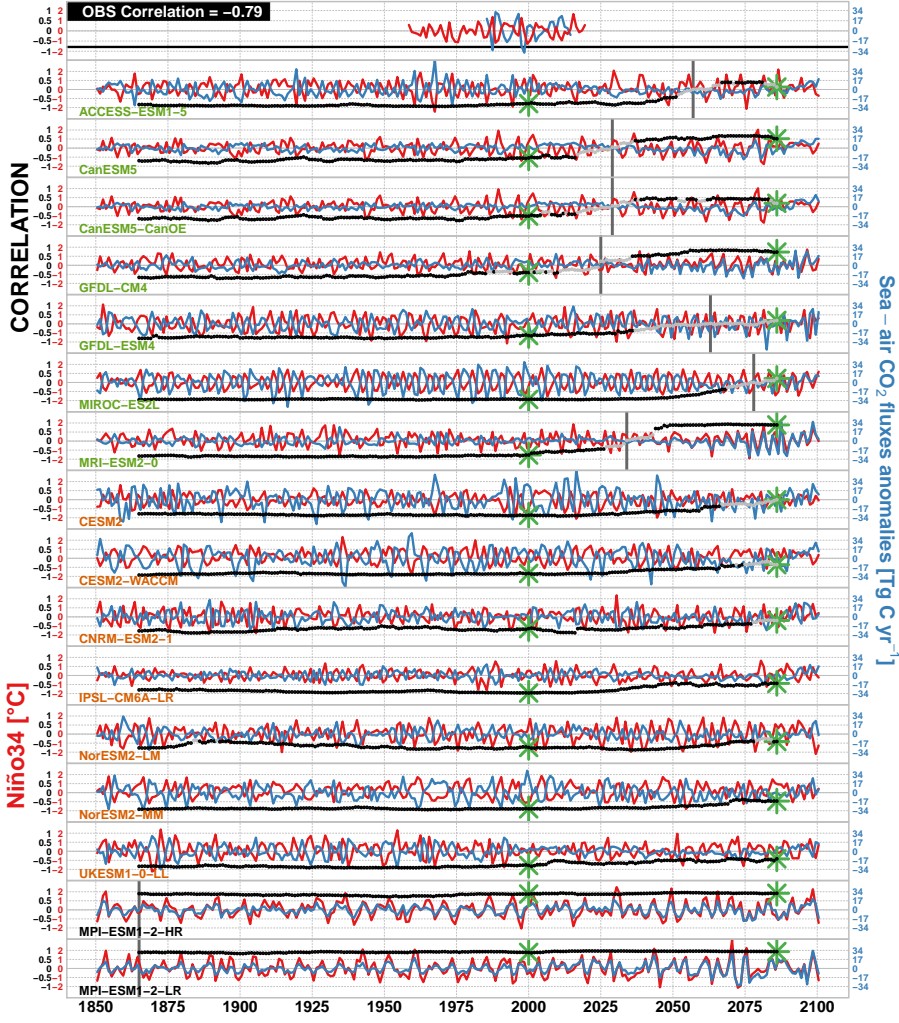

**Figure 4.** CMIP6 model ensemble annual time series of the Niño34 index (in °C, *red lines*), the average $CO_2$ flux anomalies over the EP area
(in Tg C $yr^{-1}$, *blue lines*), and the correlation for each 30-year moving window (significant correlation are indicated by *black* dots and the
non-significant ones are *grey*). The vertical bars indicate the center 30-year period with the first positive correlation. The first row shows the
observed time series of Niño34 index and average $CO_2$ fluxes anomalies over the 1985-2014. The green asterisks indicates the correlation of
the models over the observed and the 2071-2100 periods. Models names are given in green for the models with shifting correlation sign, in
orange for those maintaining the negative correlation and black for that simulating positive correlation already in 1850.



In this section, the characteristics of sea-air $CO_2$ flux variability associated with ENSO is investigated over the EP area. Figure 4 represents the annual Niño34 index and the annual average $CO_2$ flux anomalies from observations and 16 CMIP6

models. A correlation analysis between $CO_2$ flux anomaly and ENSO index is performed to study the the strength and direction of the linear relationship between these two variables. The statistical significance of these correlation is assessed by testing if the correlation follows a Student's t-distribution (with $N$-2 degrees of freedom, $N$ the number of years) at the 95% significance level. The correlation between annual $CO_2$ flux anomaly and annual ENSO index is given for the models for each 30-year sliding window over the 1850-2100 period. The observed correlation over the 1985-2014 is significantly negative ($r$=-0.79)

which is also the case for all the models for the beginning of the 1850-2100 period, except for the two MPI models. Among these models, seven maintain a negative correlation throughout the future period while seven display a shift toward a positive correlation which occurs as early as 2025. The CMIP6 models correlation over the observational period and the 2071-2100 period are indicated by the green asterisks in Fig. 4 and reported in Table 3. Figure S6 of the supplemental material gives the same figure as Fig.4 zoomed over the contemporary period.

**Table 3.** Standard deviations of sea-air $CO_2$ fluxes ($\sigma_{CO_2}$; in mol C m$^{-2}$ yr$^{-1}$) and Niño34 index ($\sigma_{Niño34}$; in °C), and their annual correlation coefficients ρ over the 1985-2014 period. In brackets are the standard deviation and correlation over the 2071-2100 period. Average Revelle Factor for each model and both periods are also given. Models in bold have significant correlation for both periods and are the ones selected as into 'reversed' and preserved' groups. † marks the models with shifting towards positive correlation. ‡ marks the models maintaining negative correlation. ∗ marks the model starting with positive correlation. Non-significant correlation are given in italic.

| | ρ | $\sigma_{CO_2}$ | $\sigma_{Niño34}$ | Revelle Factor |
|---|---|---|---|---|
| OBS | -0.79 | 17.55 | 0.69 | - |
| ACCESS-ESM1-5† | -0.78 (*0.14*) | 10.24 (7.86) | 0.72 (0.84) | 9.69 (12.56) |
| **CanESM5†** | **-0.55 (0.52)** | **5.58 (10.78)** | **0.89 (0.76)** | **9.67 (12.35)** |
| CanESM5-CanOE† | -0.52 (*0.19*) | 4.04 (9.55) | 0.89 (0.76) | 9.64 (12.28) |
| **GFDL-CM4†** | **-0.4 (0.73)** | **6.66 (13.2)** | **0.7 (0.71)** | **9.49 (12.56)** |
| GFDL-ESM4 † | -0.65 (*0.19*) | 10.33 (17.01) | 0.83 (0.86) | 9.66 (12.51) |
| MIROC-ES2L† | -0.94 (*0.22*) | 21.4 (9.41) | 0.86 (0.86) | 9.53 (12.60) |
| **MRI-ESM2-0†** | **-0.77 (0.89)** | **4.25 (17.82)** | **.66 (0.95)** | **9.21 (11.97)** |
| CESM2‡ | -0.86 (*0*) | 22.64 (15.09) | 0.86 (0.46) | 9.77 (12.42) |
| CESM2-WACCM‡ | -0.84 (*-0.35*) | 13.3 (15.17) | 0.68 (0.53) | 9.92 (12.41) |
| CNRM-ESM2-1‡ | -0.65 (*-0.2*) | 7.99 (12.75) | 0.63 (0.77) | 10.07 (12.95) |
| **IPSL-CM6A-LR‡** | **-0.97 (-0.44)** | **8.55 (8.2)** | **0.79 (0.64)** | **9.82 (12.62)** |
| NorESM2-LM‡ | -0.74 (-0.41) | 8.31 (9.92) | 0.83 (1.03) | 9.71 (12.63) |
| **NorESM2-MM‡** | **-0.89 (-0.46)** | **17.72 (10.28)** | **0.91 (0.73)** | **9.69 (12.56)** |
| **UKESM1-0-LL‡** | **-0.78 (-0.42)** | **7.28 (8.18)** | **0.64 (0.77)** | **10.04 (12.54)** |
| MPI-ESM1-2-HR∗ | 0.87 (0.92) | 7.95 (11.72) | 0.91 (0.93) | 9.55 (12.68) |
| MPI-ESM1-2-LR∗ | 0.89 (0.93) | 7.43 (17.84) | 0.87 (1.00) | 9.58 (12.66) |





Figure 4 also shows that the amplitude of $CO_2$ fluxes anomalies and their covariance with the Niño34 index are not uniform across the models. The correlation between sea-air $CO_2$ flux anomalies and Niño34 are given in Table 3 along with their respective standard deviations $\sigma_{CO_2}$ and $\sigma_{Niño34}$. The contemporary variability of $CO_2$ flux anomaly is underestimated by most of the models (see Table 3) and increases or decrease in the future according the models. Six models given in bold in Table 3 are selected to illustrate the shifting and non-shifting $CO_2$ fluxes anomalies response to ENSO variability in their future

projections. These are the models are selected because they reproduce best the observed Niño index and $CO_2$ flux anomalies correlation in the contemporary period while the correlation is significant over contemporary and future periods.

The monthly Niño34 index of the six selected models are presented against the $CO_2$ fluxes anomalies in Fig. 5, both for the contemporary (1985-2014) and future (2071-2100) periods. Values from present-day observations are also depicted. The models in the first row (CanESM5, GFDL-CM4, MRI-ESM2-0) show a change of the Niño34-$CO_2$ flux correlation while the

models in the second row (IPSL-CM6A-LR, NorESM2-MM, UKESM1-0-LL) maintain the sign of the correlation between 1850 and 2100. This reversal is thus independent on the performance of the model over the contemporary period, though the models in the first row tend to simulate lower than observed $CO_2$ flux anomaly variability. Hereafter, these first row models that simulate a reversal in ENSO-$CO_2$ flux relationship are referred to as "reversed" ESMs while the other three ESMs that maintain the contemporary relationship are referred to as "preserved" ESMs. These two groups of models are confronted in

further analysis.

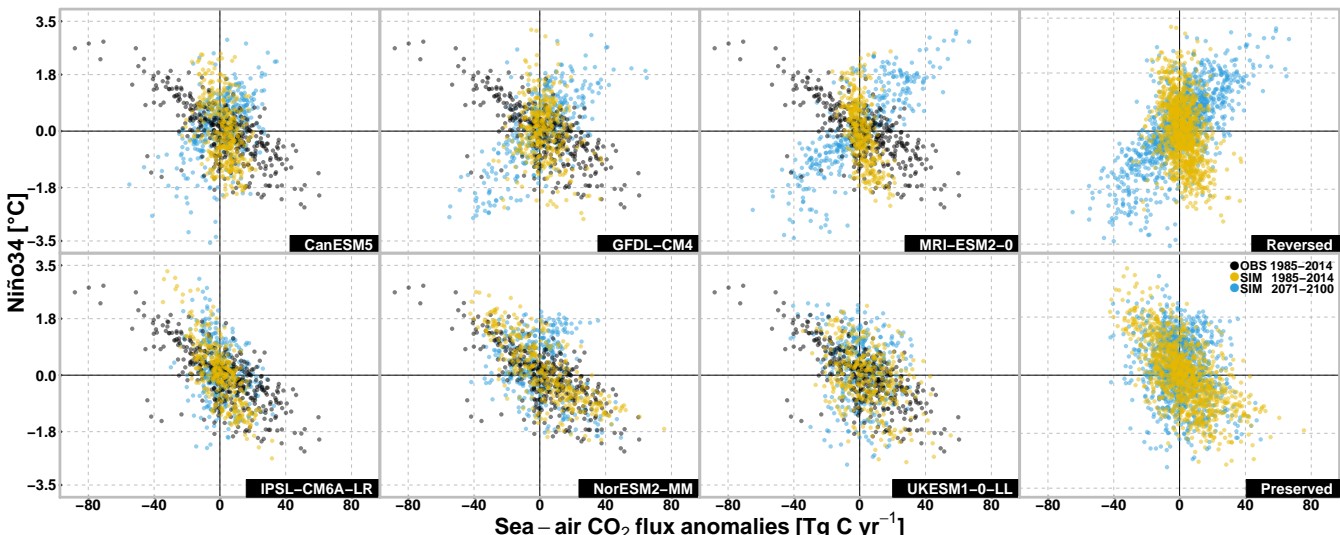

**Figure 5.** Scatter plots for the six selected models of the monthly Niño34 index (in °C) against the monthly $CO_2$ flux anomalies (in Tg C yr$^{-1}$) average over the EP domain in the 1985-2014 contemporary period (*in yellow*) and the 2071-2100 (*in blue*) period. The observed scatter plot is given in black. Top panels show CanESM5, GFDL-CM4, MRI-ESM2-0 and all reversed ESMs. Bottom panels are for IPSL-CM6A-LR, NorESM2-MM, UKESM1-0-LL and all preserved ESMs.





### 3.3 Drivers of ENSO-CO$_2$ flux variability

In order to elucidate the drivers of the modified relationship in the reversed ESMs, the thermal and non-thermal contributions to $p$CO$_2$ are investigated. Figure 6 represents the average El Niño and La Niña of $p$CO$_2$ anomalies mean for the reversed and preserved ESMs over the early historical (1851-1880), contemporary and future periods. As expected, $p$CO$_2$ thermal (non-
thermal) component always induces positive (negative) anomalies during El Niño while the opposite is true during La Niña. The non-thermal component is rather dominant (non-thermal/thermal ratios > 100%) under the early historical period (1851-1880) and even more dominant during La Niña (bigger ratios). This explains the total $p$CO$_2$ positive anomalies during La Niña (consistent with enhanced CO$_2$ outgassing; Fig. 2) and the negative anomalies during El Niño (consistent with weakened CO$_2$ outgassing) for both groups of ESMs over the early historical and contemporary periods. Over the future period, the dominance of the non-thermal component is even enhanced for preserved ESMs, which maintain the same CO$_2$ flux-ENSO relationship.
However, for the reversed ESMs the thermal component becomes dominant by the end of the 21$^{st}$ century (ratio<100%) inducing total $p$CO$_2$ negative anomalies during La Niña and positive anomalies during El Niño. The dominance of the thermal component explains the reversal in the ENSO-CO$_2$ flux relationship highlighted in Figs. 4 and 5.

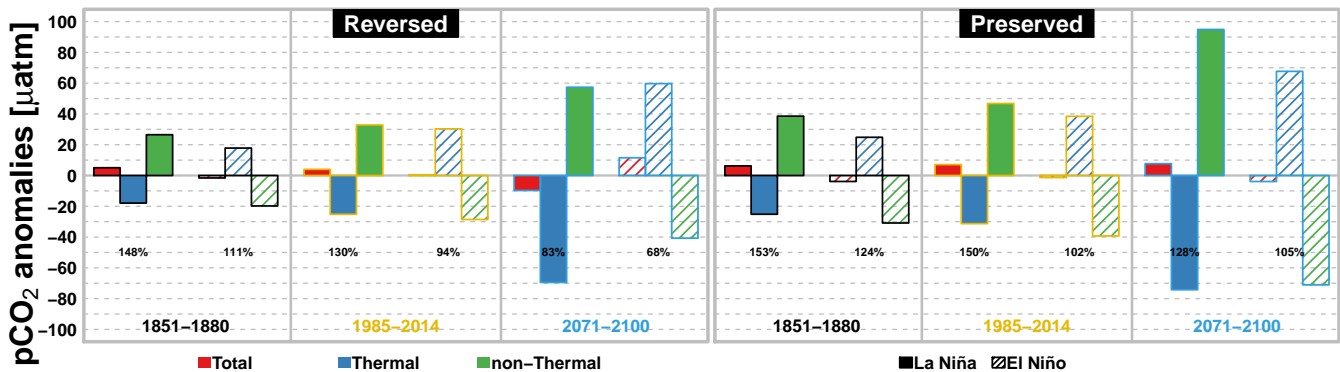

**Figure 6.** El Niño and La Niña average of total (*in red*), thermal (*in blue*) and non-thermal (*in green*) $p$CO$_2$ mean anomalies (in µatm) for the reversed (*left*) and preserved (*right*) ESMs over the early historical (1851-1880), contemporary (1985-2014 ) and future (2071-2100 ) periods in the EP domain. The absolute ratio between the non-thermal and thermal components is given (in %) for each period, group and ENSO phase.

In a high CO$_2$ future, it is expected that the $p$CO$_2$ will be more sensitive to SST and surface DIC modulations due to lower
buffering capacity (Fig. 1; *e.g.*, see also Gallego et al. (2020)). It is therefore useful to determine whether or not the reversal in the ENSO-$p$CO$_2$ response can solely be attributed to the background atmospheric CO$_2$ increase. Indeed, the non-thermal component is already dominant and will become more dominant as CO$_2$ rises. In order to test this hypothesis, the anomaly estimates of the thermal and non-thermal components of early historical ENSO $p$CO$_2$ signals are scaled to higher background $p$CO$_2$, namely contemporary and future periods. This enables us to evaluate how the non-thermal/thermal ratio varies into the
future assuming no change in the biological and physical forcing (i.e. amplitude of ENSO-induced changes in SST and DIC



are unchanged). This is done by keeping the dSST variable in Eqs. 1 and 2 at early historical period, while scaling up the $pCO_2$ elements to contemporary and future values. A similar figure as Fig. 6 showing these scaled components is given in Fig. S7 of the supplementary material. Following this scaling, the non-thermal component remains dominant for the three periods in both groups of models. This means that the $pCO_2$ increase alone cannot explain the reversal behaviour in the reversed ESMs. It

suggests changes in biological and physical forcing are also responsible for the thermal component becoming more dominant in this group of ESMs.

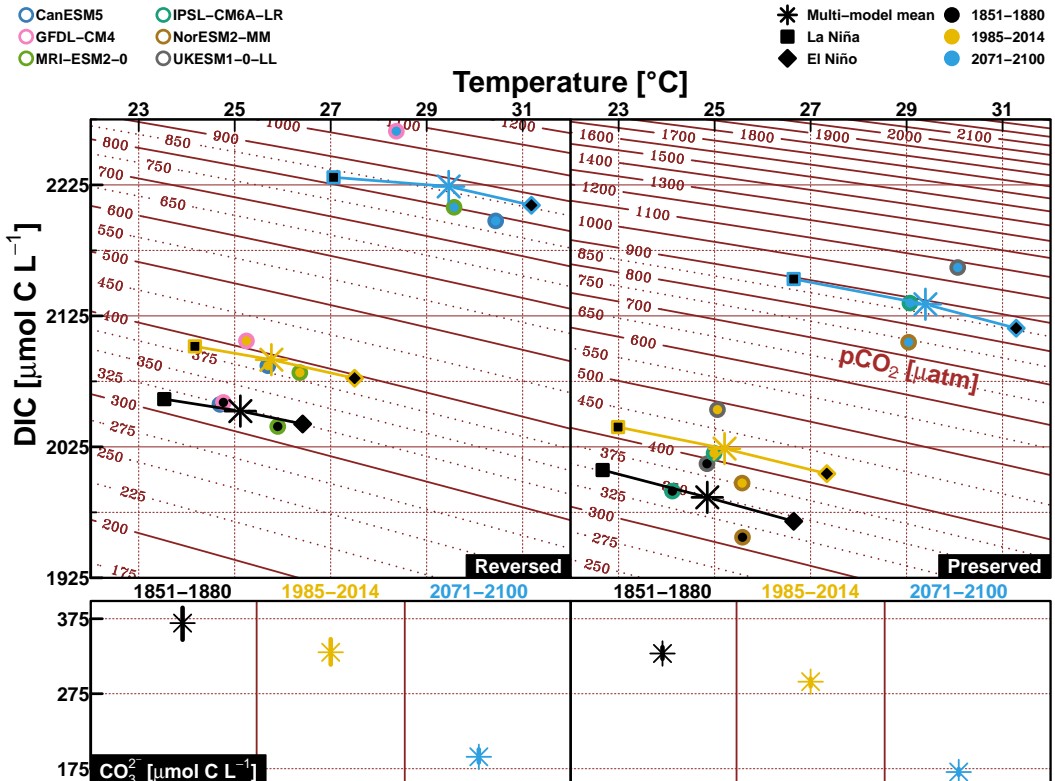

**Figure 7.** Mean SST (in °C) versus mean DIC (in μmol C L$^{-1}$) over the early historical (1851-1880), contemporary (1985-2014) and future (2071-2100) periods in the EP domain simulated by all reversed and preserved ESMs (*top* panels, circle markers). The multi-model mean values of SST and DIC (asterisk markers) from each ESM group together with their respective mean values during La Niña (square markers) and El Niño (diamond markers) are also depicted for the three periods. Isolevels of $pCO_2$ for varying SST and DIC are given in the background. Bottom panels show the multi-model range and mean of surface carbonate concentration (in μmol C L$^{-1}$) for both groups and three periods.

Next, we quantify the $pCO_2$ sensitivity to ENSO-induced temperature and DIC changes across different time periods. Figure 7 shows the mean states of SST against surface DIC for reversed and preserved ESMs over the early historical, contemporary and future periods. $pCO_2$ isolevels for varying SST and DIC are computed using the carbonate system parameters codes from

the R package "seacarb" (Gattuso et al., 2020). These values have been computed using surface alkalinity from multi-model



mean state (over the 1850-2100 period) from reversed and preserved groups separately. The multi-model range and mean of average surface carbonate ion concentration is also given for both groups over the three periods.

All models show higher sensitivity of $p\mathrm{CO_2}$ to temperature and DIC perturbations in the future, *i.e.* the same variations of DIC or temperature in the future will induce a stronger change in surface $p\mathrm{CO_2}$. Indeed, $p\mathrm{CO_2}$ isolevels are getting closer as

SST and DIC increase (see Fig.7). The main difference between the two groups is that the reversed models simulate (i) higher surface DIC increase from early historical or contemporary to future periods and (ii) lower range of DIC changes during ENSO phases. The $p\mathrm{CO_2}$ level and its increase of across different time periods are very similar between the two ESM groups. The simulated temperature changes are also similar between both groups. The higher surface DIC increase in the reversed models can be explained by the higher $\mathrm{CO_3^{2-}}$ ion concentration at beginning of the transient simulation, which translates to higher

carbon buffer capacity and allow these models to take up more excess carbon from the atmosphere. The lower surface DIC range (between in La Niña and El Niño regimes) in the reversed models could be associated with changes in biology- and/or upwelling-induced surface DIC fluctuations.

Even without climate change, the influence of biological production on perturbing surface $p\mathrm{CO_2}$ is expected to increase with higher Revelle factor in the future. Here, we quantify the contribution of biological production in reducing the surface

$p\mathrm{CO_2}$ (*i.e.*, $\Delta p\mathrm{CO_2}_{\ \mathrm{bio}}$) during both La Niña and El Niño phases according to Eq. 3. In the contemporary period, stronger primary productivity during La Niña attenuates the upwelling-induced $p\mathrm{CO_2}$ increase, and vice versa during El Niño. In addition, this anomaly pattern observed in the contemporary period is maintained into the future (see Fig. S8 of the supplemental material depicting time-series of the average intPP computed over the EP area). Figure 8 shows that these biological contributions significantly increase in the future, with higher $\Delta p\mathrm{CO_2}_{\ \mathrm{bio}}$ persists during La Niña phase. This stronger contrast in

biologically-induced $\Delta p\mathrm{CO_2}_{\ \mathrm{bio}}$ difference between La Niña and El Niño regimes is also enhanced by the increased future primary production variability simulated in the respective ESMs (Fig. S8). The projected variability in primary production between La Niña than El Niño is even bigger for the reversed than preserved ESMs (i.e., by up to a factor of five larger; see Fig. S8). Note that the majority of the chosen ESMs simulate a declining trend in the primary production toward the end of the 21$^{\mathrm{st}}$ century. The export production at 100m also shows similar ENSO-induced variability and evolution as the intPP (not

shown).

As stated above, the primary reason for the enhanced biological contribution on $\Delta p\mathrm{CO_2}_{\ \mathrm{bio}}$ is driven by the increasing Revelle Factor with higher atmospheric $\mathrm{CO_2}$ concentrations in the future (see Fig. 1 and Hauck and Völker, 2015). Assuming that the upwelling-induced DIC variation stays constant in the reversed ESMs, an enhanced primary production fluctuation (higher during La Niña, lower during El Niño) in the future would decrease the ratio between non-thermal and thermal $p\mathrm{CO_2}$

components and therefore could contribute to the simulated reversed relationship (Fig. 6). Fig. S8 also shows that the preserved ESMs also simulate enhanced primary production variability but with a lesser magnitude than the reversed ESMs. Yet the contemporary ENSO-$\mathrm{CO_2}$ flux relationships in this ESM group are maintained in the future, suggesting too low biological contribution or other additional processes are at play.

In addition to surface biological activities, the reduction of the non-thermal contribution to the total $p\mathrm{CO_2}$ in the reversed

ESMs can also be attributed by changes in upwelling-induced surface DIC modulation. Here, we examine the mean vertical

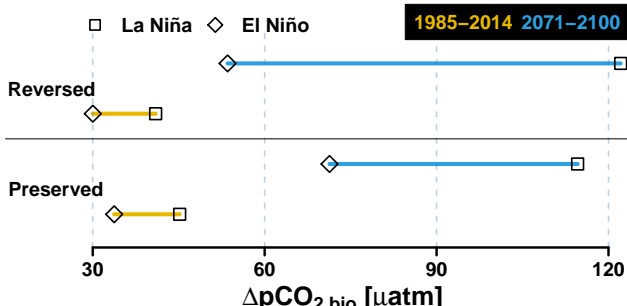

**Figure 8.** Multi-model mean average biological contribution to the oceanic $pCO_2$ (in µatm) deficit during La Niña and El Niño regimes for the 1985-2014 and 2071-2100 period for all reversed (*top*) and preserved (*bottom*) models in the EP domain.

profiles of DIC and temperature and carbonate ion in the EP domain across the two ESM groups. Figure 9 shows the average vertical profiles of DIC and temperature for the two groups of ESMs over the the EP domain from the surface down to 300 m depth. Both groups consistently show DIC and temperature increase in the future, but the change varies in magnitude and vertical distribution.

Indeed, the reversed ESMs simulate higher historical DIC (yellow lines in first row of Fig. 9) making them more biased than the preserved ones. The simulated DIC increase is similar at 100 m and deeper for both groups (purple dashed lines). However, the increase from the surface to 100 m is larger for the reversed ESMs. This leads to a stronger reduction in vertical DIC gradient, which would also contributes to a less ENSO-induced surface DIC variability in the reversed ESMs. This is also consistent with the projected more dominant thermal contribution relative to the total $pCO_2$. The future increase in the
upper ocean DIC concentration is associated with the uptake of anthropogenic carbon from the atmosphere. We note that the increase in DIC concentration at depth can also be associated with the shallow water overturning circulation, which advects southern DIC-rich (and carbonate poor) waters into the region (Toyama et al., 2017; Rodgers et al., 2020) and can also affect the buffering capacity of upwelled watermass.

The higher surface DIC increase is also illustrated in the right panel Fig. 10, depicting that the reversed ESMs simulate more
carbon uptakes (or less cumulated DIC loss because the tropical Pacific is a mean outgassing system) than the preserved models over the transient simulation period. This is attributed to the higher $CO_3^{2-}$ concentration simulated by the reversed ESMs at the beginning of the transient simulation from surface to 300m depth (see bottom panels of Fig. 9 and left panel of Figure 10 for surface $CO_3^{2-}$). Hence, reversed ESMs have higher buffering capacity which makes them able to uptake more atmospheric carbon. This is the first order explanation for the projected higher surface $CO_3^{2-}$ reduction (see bottom panels of Fig. 9 and
middle panel of Figure 10).

The relationship between historical surface carbonate concentration and $CO_2$ uptakes can be generalised for all models. Figure 11 shows contemporary surface carbonate concentration against the cumulated sea-air $CO_2$ flux from 1850 to 2100 over EP for all the models except the MPI models. The correlation at 0.67 indicates that the carbonate concentration is a good indicator of the buffering capacity of the model: the higher the carbonate the lower the cumulated $CO_2$ outgassing (ie. more





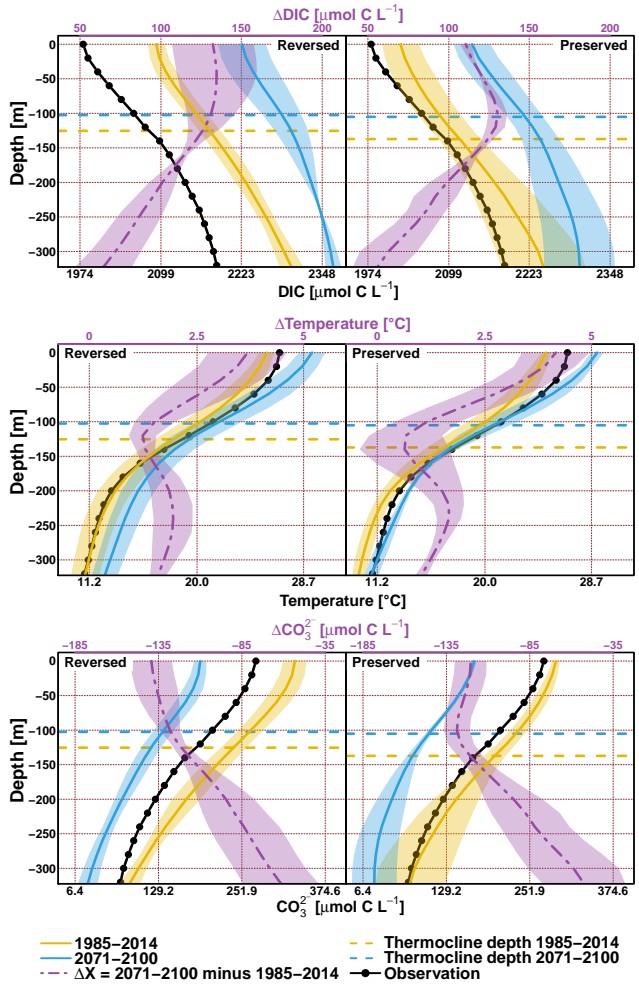

**Figure 9.** Multi-model mean of vertical DIC (in µmol C L$^{-1}$, top panels), temperature (in °C, middle panels) and carbonate ion concentration (in µmol C L$^{-1}$, bottom panels) profiles over the 1985-2014 (*in yellow* lines) and 2071-2100 (*in blue* lines) periods for reversed (*left*) and preserved (*left*) models. The profile difference between both period profile (Δ) is given *in purple* dashed-dotted lines. The black lines with dots are the observed profile for the three variables. The dashed horizontal lines indicate the average thermocline depth for each groups and time periods. One standard deviation is given in shaded colours.

carbon uptakes). The preserved ESMs are less biased in terms of carbonate concentration, which tend to indicate that their behaviour should be more reliable.

The preserved ESMs simulate stronger warming at the surface (see middle panels of Fig. 9), suggesting stronger future stratification, which is consistent with the higher increase in the subsurface DIC (e.g., associated with the biological remineralisation) with less upwelling. Consequently, the weaker future stratification in the reversed ESMs is also consistent with the

more uniform DIC vertical profile.





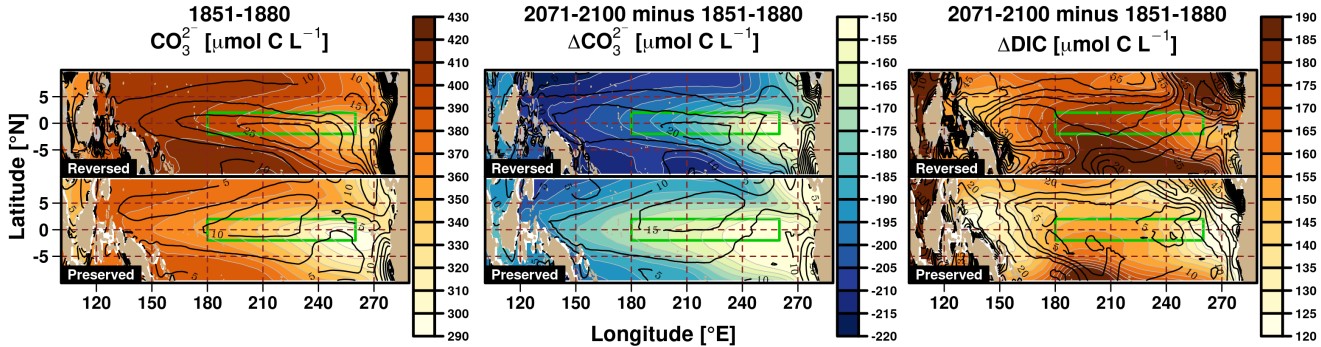

**Figure 10.** Maps of average surface $CO_3^{2-}$ concentration (*left* in µmol C L$^{-1}$) for the reversed (*top*) and preserved ESMs for the 1851-1880 period. The middle column shows the carbonate ion concentration difference between the 2071-2100 and 1851-1880 periods. The right column show the surface DIC concentration difference between the 2071-2100 and 1851-1880 periods. The green boxes outline the EP region.

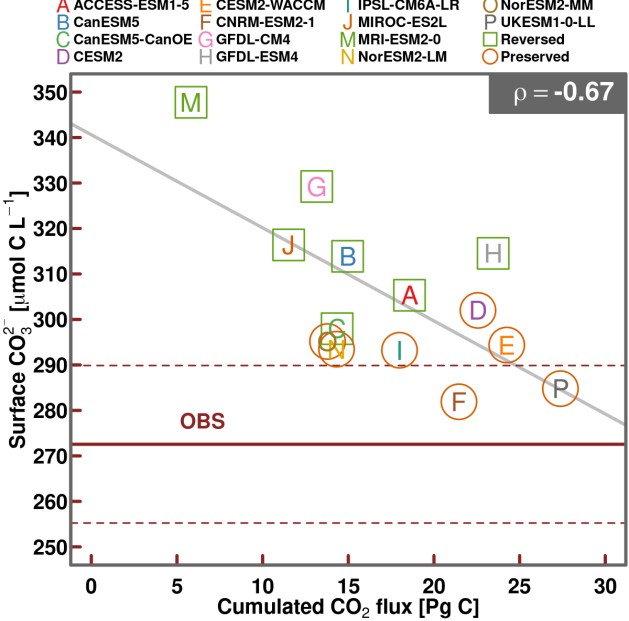

**Figure 11.** Average contemporary surface $CO_3^{2-}$ concentration (in µmol C L$^{-1}$) plotted against the cumulated sea-air $CO_2$ fluxes (in Pg C) from 1850 to 2100 in the EP region. ρ is the correlation and green and orange square respectively indicates the reversed and perserved ESMs.

ENSO-induced upwelling variability alters the surface DIC anomalies. However, there is no significant difference in the thermocline depth evolution between the reversed and preserved ESM groups. The thermocline depths are expected to become shallower toward the end of the 21$^{st}$ century, consistent with future warmer upper layer and stronger stratification. In all ESMs, the thermocline depth variation due to ENSO, *i.e.* shallower thermocline depth during El Niño events (indicating the



anomalously weak upwelling state) and vice versa during La Niña, is maintained in the future. Figure S9 of the supplemental material depicts time-series of the average thermocline depth computed over the EP domain. Despite future shallowing of thermocline depth, the ENSO-driven surface DIC variation in all ESMs (anomalously lower DIC during El Niño and higher DIC during LA Niña) is also maintained in the future (see Fig. S10).

## 4  Discussion, limitations and perspectives

In the tropical Pacific, the dominant mode of sea-air $CO_2$ fluxes variability over the interannual time scale has been established to be associated with ENSO. Here, by evaluating the capacity of 16 CMIP6 ESMs to reproduce this relationship over the historical period provides a valuable means to validate their performance. As shown in Table 3, while most ESMs are able to reproduce the observed contemporary relationships (i.e., negative correlation or outgassing anomaly during La Niña and vice versa during El Niño), there are two ESMs that simulate the complete opposite relationship. Furthermore, the amplitude of
the Niño34 ($CO_2$ fluxes) variability also varies considerably among models over the contemporary period, from 0.91 (0.23) to 1.32 (1.29) times, as compared to the observations (Table 3). As with previous generation ESMs (Jin et al., 2019), considerable differences in the spatial extent of $CO_2$ flux anomaly patterns associated with ENSO variability are also simulated in the current CMIP6 ESMs.

Model projections suggest an enhanced ENSO variability in the future associated with the intensification of upper-ocean
stratification (Cai et al., 2018). Due to the climate-carbon cycle feedback, analysing how the ENSO-induced $CO_2$ fluxes will be altered by future climate change could provide a valuable insight on the projections of long-term anthropogenic climate change (Betts et al., 2020). Among the analysed ESMs, half of models show a reversal in their ENSO-$CO_2$ flux relationship in the Equatorial Pacific (i.e., from an anomalous $CO_2$ uptake to outgassing during El Niño and vice versa during La Niña events) under the strongest future climate change scenario SSP5-8.5. This reversed relationship, superimposed on the projected
ENSO-$CO_2$ fluxes by the land biosphere (Kim et al., 2016), suggests an even stronger increase in atmospheric $CO_2$ growth rate during future El Niño events. Nevertheless, our assessment indicates that ESMs that simulate this reversed pattern also simulate considerable bias in the contemporary surface $CO_3^{2-}$ concentration; therefore, the projections from these ESMs should be considered with caution.

The readers must keep two things in mind while interpreting the results of this study: (i) only the high emissions SSP5-8.5
scenario has been considered. Results may be scenario dependent, especially with respect to the future atmospheric $CO_2$ level. (ii) The models have been grouped (and averaged) into two categories to identify patterns or consistencies and to simplify the analyses. In addition, we have focused our analysis in contrasting El Niño and La Niña for a confined region in the Equatorial Pacific (i.e., 2°S-2°N and 180°-260°E). We have also applied our analysis on a slightly larger domain (5°N-5°S, 150°W-240°W), and the overall conclusions remain consistent (not shown).

Accurate representations of the tropical Pacific mean climate state and ENSO-related hydrographic changes in models are fundamental for ENSO impact studies. For instance, ESM projections of precipitation changes associated with future ENSO depend on the contemporary SST biases and trends (Stevenson et al., 2021). Simulating the contemporary tropical Pacific



climate accurately has been a great challenge for the modelling community over the past decades, but evidence of continuous improvements over preceding generation ESMs is a promising sign (*e.g.,* IPCC, 2021; Bellenger et al., 2014).

We show that the simulated amplitude and spatial extent of physical and biogeochemical properties induced by ENSO vary considerably across ESMs. Future model development should therefore focus on capturing the observed mean state as well as the regional anomalies pattern during dominant climate modes, such as La Niña vs El Niño phases. To achieve these developments, long-term interior carbon chemistry observations are needed. In particular, vertical distribution of DIC/ALK/CO$_3$ concentrations during El Niño and La Niña would be extremely helpful to constrain the contrasting ESM projections.

The important roles of vertical DIC gradient and biological production in the reversal of the ENSO-CO$_2$ flux relationship are also highlighted in this study. For example, the increased primary production variability that contribute the reversed ENSO-CO$_2$ flux relationship can be associated with model-dependent primary production formulation (e.g., sensitivity of phytoplankton growth rate to temperature) and circulation-driven nutrient upwelling patterns, among others. We note that elucidating the drivers of enhanced primary production in each ESM is beyond the scope of this paper.

Future model developments are also necessary to ensure that ESMs are able to reliably capture multiple layers of non-linear processes that connect ENSO variability and sea-air CO$_2$ fluxes in the Equatorial Pacific. The latest generation of ESMs have progressed considerably in reproducing key climatological properties of surface ocean biogeochemistry (Séférian et al., 2020). Future advancements could focus on improving the biogeochemical representation in the interior as well as better understanding of the physical-biogeochemical interactions across various time scales, as well as across different regions. For instance, outside

the tropical Pacific, the ocean carbon cycle are modulated by different climate modes, such as the North Atlantic Oscillation (Keller et al., 2012; Tjiputra et al., 2012) and the Southern Annular Mode (Lenton and Matear, 2007; Keppler and Landschützer, 2019). Future studies that advance our understanding of how the ocean carbon cycle in these regions might be affected by future anthropogenic climate change could be valuable to further reduce uncertainties in future climate projections.

## 5   Summary

In this paper, the ENSO-induced response of sea-air CO$_2$ fluxes under a high CO$_2$ future climate scenario is presented using observed data and model simulations from CMIP6 ESMs. The heart of the work was to examine the roles of two concurrent physical and biogeochemical processes driving the sea-air CO$_2$ fluxes variability: (i) anomalously high (low) surface temperature that leads to low (high) CO$_2$ solubility, which enhances (reduces) outgassing, and (ii) anomalously strong (weak) upwelling that brings more (less) DIC-rich water to the surface and enhances (reduces) outgassing. Opposing effect of these two processes

is enhanced by ENSO: high sea surface temperature is associated with weaker upwelling and stronger stratification during El Niño and the opposite occurs during La Niña.

The findings can be summarised as following:

–   During the historical period, observational data shows that sea-air CO$_2$ flux anomalies are negatively correlated with ENSO-associated warming, and this is reproduced in the vast majority of the models (14 of 16);



– Under the high emissions future projection (SSP5-8.5), this correlation persists in half of the examined models (7 of 14),
        but is projected to reverse across the other half;

       – Depending on the model, the future variability of $CO_2$ fluxes anomaly in the Equatorial Pacific domain could either
        increase or decreases. This is consistent with the projected $pCO_2$ variability over the same area (Gallego et al., 2020).
        However, Liao et al. (2021) found weaker future $CO_2$ flux anomalies during ENSO phases which maybe partly related
to their model selection in their analyses.

       – All the models shows a higher Revelle Factor in the future, leading to a stronger $pCO_2$ sensitivity to changes in surface
        temperature between ENSO phases (similar results has been shown for CMIP5 Gallego et al., 2020);

       – In this study, the mechanisms leading to the reversal of this ENSO-$CO_2$ flux relationship are explained by the thermal
        contribution to $pCO_2$ becoming more dominant relative to the non-thermal component. This is explained by (i) the
increase in the $pCO_2$, (ii) the enhanced primary production fluctuation, and (iii) the upper ocean DIC concentration
        increase (due to increasing anthropogenic $CO_2$ uptake) which decreases the vertical gradient in the thermocline, and
        eventually attenuating the ENSO-modulated surface DIC variability;

       – A reversing ENSO-$CO_2$ flux relationship over the 21st century projected in some ESMs seems unlikely since it is a direct
        consequence of a strong bias in the mean state of carbonate ion concentration over the historical period.

*Data availability.* The neural-network-based interpolated $CO_2$ product used in this study is freely accessible at the National Centers for
       Environmental Information via https://www.ncei.noaa.gov/access/ocean-carbon-data-system/oceans/SPCO2_1982_2015_ETH_SOM_FFN.
       html. The Japanese 55-year reanalysis SST product used in this study is accessible from their Web site at search.diasjp.net/en/dataset/JRA55.
       The vertical temperature, DIC (climatology) and ALK are respectively available at https://icdc.cen.uni-hamburg.de/daten/reanalysis-ocean/
       easy-init-ocean/ecmwf-oras5.html, https://www.ncei.noaa.gov/access/ocean-carbon-data-system/oceans/ndp_104/ndp104.html and https://
www.glodap.info/index.php/mapped-data-product/. The CMIP6 data used in the analysis were obtained from https://esgf-node.llnl.gov/
       search/cmip6.

*Author contributions.* PV and JT conceived the study. PV prepared the data and figures, conducted the analysis and wrote the original
       manuscript, JT contributed to improving the methodology and the analysis and interpretation and editing the manuscript. All authors dis-
       cussed, commented and edited the manuscript.

*Competing interests.* The authors declare no competing interest



*Acknowledgements.* All computations and figures are made using the R free software (R Core Team, 2016). PV and JT acknowledge funding from the Research Council of Norway (COLUMBIA-275268, CE2COAST-318477, and EASMO-322912).

RS particularly acknowledges the support of the team in charge of the CNRM-CM climate model. Supercomputing time was provided by the Météo-France/DSI supercomputing center. Simulation performed by CNRM-ESM2-1 was supported by the European Union's Horizon 2020 research and innovation program with the H2020 project CRESCENDO under the grant agreement No 641816 and the H2020 ESM2025 under the grant agreement No 101003536). TI was supported by the European Union's H2020 research and innovation programme under the grant agreements No. 101003536 (ESM2025), No. 821003 (4C), and No. 820989 (COMFORT), as well as by the Deutsche Forschungsgemeinschaft (DFG, German Research Foundation) under Germany's Excellence Strategy - EXC 2037 'Climate, Climatic Change, and Society' - Project Number: 390683824.



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
