# Peer review of "Contrasting projection of the ENSO-driven CO2 flux variability in the Equatorial Pacific under high warming scenario"

_Earth System Dynamics, 2022_

## Author Response (AR1)

"Contrasting projection of the ENSO-driven $CO_2$ flux variability in the Equatorial Pacific under high warming scenario" by P. Vaittinada Ayar et al.

We first would like to thank the anonymous reviewer for her/his thorough reading and very positive and constructive comments. We tried to take them into account as much as possible. A detailed point-by-point reply to these comments is provided below. Changes in the manuscript are indicated in blue.

**Answer to Referee #1 :**

**General comments :**

- *Since the air-sea CO2 flux is related to three terms : ocean pCO2, air pCO2, and wind-solubility coefficient, the authors only analyze the ocean pCO2 in the manuscript. I am very interested in the role of wind-solubility coefficient and air pCO2 in explaining the divergence in two groups of ESMs. Different models might have different wind and temperature variability which might contribute to the CO2 flux variability. It is worth quantifying and discussing the wind and solubility terms. All the ESMs might use the same air pCO2, so the air pCO2 might have a very little contribution. However, it is needed to be at least discussed in the manuscript.*

**Authors' response :** Thank you for this interesting point. We computed $k$ the gas transfer velocity multiplied by $K_0$ the solubility coefficient used to estimate $CO_2$ fluxes as F = $k * K_0$ *($pCO_{2o}$ - $pCO_{2a}$). Figure R1 represents the $k * K_0$ and surface wind anomalies for each period, group of models and ENSO phase. All the three preserved and reserved groups are able to reproduce the observed weakening in trade winds during El Niño, and vice versa during La Niña. The amplitude between ENSO phases is larger for the preserved models than the reversed ones. This can explain the higher amplitude variation between ENSO phase for the preserved models than the reverse ones (see Table 3 and Fig 5 of the article). However, for the respective groups the amplitude between ENSO phases is not changing between given the periods. This means that the wind variability can only have a marginal contribution to $CO_2$ fluxes variability and can not explain the behaviour of the reversed group models.

Figure R1 has been added to the supplementary material and this question has been addressed in lines 243-251 of the revised manuscript as :

In addition to surface ocean $pCO_2$, $CO_2$ flux is estimated using atmospheric $pCO_2$ and wind solubility coefficient $k * K_0$ as :

$$\text{fgco2} = k * K_0 * (pCO_{2o} - pCO_{2a}) \tag{4}$$

$k$ represents the gas transfer velocity and $K_0$ the solubility coefficient [cf. WANNINKHOF, 2014 ]. The anomalies of surface wind and product of $k * K_0$ for each period, group of models and ENSO phase are depicted in Fig. S8 of the supplementary material. The amplitude of both anomalies

[Figure]

FIGURE R1 – El Niño and La Niña surface wind (*top* in m s$^{-1}$) and $k*K_0$ (*bottom* in mol C m$^{-2}$ yr$^{-2}$ atm$^{-1}$) mean anomalies for the reversed (*left*) and preserved (*right*) ESMs over the early historical (1851-1880), contemporary (1985-2014) and future (2071-2100) periods in the EP domain. Vertical bars represents ± one s.d. of the anomalies for the respective periods, groups of models and ENSO phases.

between ENSO phases is larger for the preserved models than the reversed ones, which partly explains the higher amplitude of $CO_2$ flux variability variation between ENSO phase for the preserved models than the reverse ones (see Table 3 and Fig. 5). However, for the respective groups the amplitudes between ENSO phases are not changing between given the analysed periods. This means that the wind variability can only have a marginal contribution to $CO_2$ fluxes variability and can not explain the behaviour of the reversed group models.

■ *Ocean pCO$_2$ is sensitive to four terms : temperature, DIC, alkalinity, and salinity (Takahashi et al., 1993). The authors only discuss the temperature and DIC. Although the DIC is dominant in the ocean pCO$_2$ variability, the alkalinity has a very large compensation. The alkalinity might partly contribute to the model divergence. In addition, the precipitation probably changes a lot under global warming (Cai et al., 2015), this might drive a relatively large variability of alkalinity and salinity in the future. It would be convincing if the author could discuss/quantify the contribution of alkalinity and salinity to the model divergence ?*

**Authors' response :** Thank you this comment. In order to address this, similar figure as Figure 7 of the article has been reproduced for SSS vs. salinity normalised ALK (ALKs). Except, this time

we did one figure by group and period given the important increase of temperature and DIC from early historical to the future (cf. revised Figure 7 of the article). This figure shows a higher SSS and ALKs for reversed models than the preserved ones. For both groups, the ALKs and SSS changes are very small from one period to another indicating a limited sensitivity of pCO$_2$ to salinity and alkalinity. Besides, the amplitude between ENSO phases is small for salinity and alkalinity (respectively $<.2$ psu for SSS and $< 6.5$ µmol eq L$^{-1}$ for ALKs).

[Figure]

FIGURE R2 – Mean SSS (in psu) versus mean ALKs (in µmol eq L$^{-1}$) over the early historical (1851-1880), contemporary (1985-2014) and future (2071-2100) periods in the EP domain simulated by all reversed and preserved ESMs (*top* panels, circle markers). The multi-model mean values of SSS and ALK (asterisk markers) from each ESM group together with their respective mean values during La Niña (square markers) and El Niño (diamond markers) are also depicted for the three periods. Isolevels of pCO$_2$ for varying SSS and ALKs are given in the background and are computed from period and group specific SST and DICs muti-model average.

Figure R2 has been added to the supplementary material and this has been addressed in lines 276-283 of the revised manuscript as :

TAKAHASHI et al., 1993 also mention pCO$_2$ sensitivity to alkalinity and salinity. A similar figure as Fig. 7 but for the mean states of SSS against ALKs is given in supplementary Fig. S10. Given the important increase of temperature and DIC from early historical to the future period (cf. Fig. 7) one panel per model group and period is produced. This figure shows higher SSS and ALKs for reversed models than the preserved ones. For both groups, the ALKs and SSS changes are very small from one period to another indicating a limited sensitivity of pCO$_2$ to future changes in salinity and alkalinity. Besides, the amplitude between ENSO phases is small for salinity and alkalinity (respectively $<.2$ psu for SSS and $< 6.5$ µmol eq L$^{-1}$ for ALKs). Therefore the relative contribution of ENSO-induced salinity and alkalinity changes to pCO$_2$ is smaller than temperature and DIC changes.

■ *Line 245-247. The authors found two differences between two group of ESMs (Large increase of surface DIC and lower range of DIC changes). Fig.9 could show the large increase in surface DIC. However, I could not see a figure showing the lower range of DIC changes during ENSO phases. I would suggest one such figure in the main text or supporting information.*

**Authors' response :** Figure 7 has been modified replacing DIC by salinity normalised DIC (DICs) to better see the DIC changes during ENSO phases. The range of DIC changes during ENSO phase goes from 30.26 to 19.45 µmol C L$^{-1}$ for reversed models and from 44.87 to 34.00 µmol C L$^{-1}$ for preserved ones.

This has been adressed in lines 268-269 of the revised manuscript as :
(from early historical to future period, the absolute change of surface DICs between both ENSO phases evolves from 30.26 to 19.45 µmol C L$^{-1}$ for reversed models and from 44.87 to 34.00 µmol C L$^{-1}$ for preserved ones)

■ *Fig. 9. Except for surface DIC difference between preserved and reversed models, I also see the difference in subsurface DIC (e.g., 200-300 m) between two groups of ESMs. What is the role of subsurface DIC difference in the model CO2 flux-ENSO relationship divergence ? Why is the subsurface DIC also different in the two groups of models ?*

**Authors' response :** Vertical DIC gradient is a key factor driving ENSO related $CO_2$ flux variability throughout the vertical column. The reversed ESMs simulate higher historical DIC making them more biased than the preserved ones, but both groups have similar vertical profile. As stated in the article, the increase of future DIC below 100m is similar in both groups. But we noted that there is a higher DIC increase in the upper ocean for the reversed ESMs, leading leading to a stronger reduction in vertical DIC gradient and thus contributes to a less ENSO-induced surface DIC variability in the reversed ESMs. The difference in the subsurface DIC between both groups is likely associated with the bias in the respective alkalinity mean state (see also new supplemental figure S11). Model-dependent bias in interior alkalinity can arise from the different formulations of particulate inorganic carbon formation and dissolution in each model (Planchat et al., in prep.).

This has been adressed in lines 311-315 of the revised manuscript as :
Indeed, vertical DIC gradient is a key factor driving ENSO related $CO_2$ flux variability throughout the vertical column. The reversed ESMs simulate higher historical DIC (yellow lines in first row of Fig. 9) making them more biased than the preserved ones, but both groups have similar vertical profile. Bias in the interior DIC may be associated with the simulated mean alkalinity state (Supplementary Fig. S9 and S11), which is likely associated with variation in particulate inorganic carbon formulation in ESMs (Planchat et al., in prep., 2022).

**Minor comments :**

■ *Line 41-43. The tropical Pacific ocean CO2 flux anomaly is not only related to the upwelling strength but also related to the poleward Ekman transport driven by easterly trade wind. One more*

*reference (Liao et al., 2020 GBC) is suggested.*

**Authors' response :** This has been added in lines 45-46 of the revised manuscript as :
In addition, $CO_2$ flux anomaly variability in the tropical Pacific is also related to the poleward Ekman transport driven by the easterly trade wind [LIAO et al., 2020].

- *Line 80. What is the re-grid method ?*
  **Authors' response :** Bilinear interpolation is used as re-griding method. This has been clarified in line 85 of the revised manuscript.

- *Line 91. The ENSO variability is usually an interannual variability ranging from 3 to 7 years. Could the author plot the total CO2 flux and CO2 flux anomaly at a sample point to show how well is the detrend method ? Could the detrend method remove the decadal variability ?*

[Figure]

FIGURE R3 – Example of sea-air $CO_2$ flux (in mol C m$^{-2}$ yr$^{-1}$, *bottom*) total, anomaly and trend extracted at a NorESM1-MM grid-point.

**Authors' response :** From the example in Figure. R3, it shows that our detrending method remove the long-term (greater than multi-decadal) trend and preserve both interannual and decadal variability.

- *Line 100. Why do you use 1981-2010 as the climatological period instead of 1985-2014 which is the contemporary period defined in the manuscript.*

  **Authors' response :** The 1981-2010 climatology has been chosen because this is the usual climatological period chosen by institute providing Niño3.4 index. See for instance,
  https ://psl.noaa.gov/gcos_wgsp/Timeseries/Nino34/
  https ://climate.copernicus.eu/charts/c3s_seasonal/c3s_seasonal_plume_lfpw ?facets=undefined&time =2022050100,0,2022050100&type=plume&area=nino34.
  The 1985-2014 is chosen given to availibility of $CO_2$ flux observation (1982-2015).

- *2 Caption. What is the observed data of SST and CO2 flux ? I know the authors state them in the method section. However, it would be clearer for the readers if the authors could detail them in the caption. For example, SST is JRA.*
  **Authors' response :** Done.

- *3. Why do the authors use 5N-5S instead of 2N-2S ? This is not consistent with the method section.*

   **Authors' response :** That was a typo, thank you for noticing this. It is $\pm 2°$ and we have corrected this in the revision.

- *What is the CMIP6 ensemble anomalies one standard deviation ?*

   **Authors' response :** Fig. 2 and 3, it represents the CMIP6 inter model ensemble standard deviation of the different variables during El Niño and La Niña phase.

- *Line 379. The text reads like Liao et al. (2021) selected the model subjectively and got a partial and biased conclusion. Actually, Liao et al., (2021) use a strict and reasonable constrain method to select the model. The results are physically rational and convincing. I would suggest rephrasing the words. A suggested way would be : "With a strict constrain method based on contemporary observations, the model tends to show a weaker future CO2 flux anomalies during ENSO phases (Liao et al., 2021)."*

   This has been rephrased in lines 417-418 of the revised manuscript as :
   However, using models selected based on their contemporary period performances, Liao et al., 2021 found weaker future $CO_2$ flux anomalies during ENSO phases.

- *Lines 381-382. The increasing Revelle factor and ocean pCO2 sensitivity to temperature would be a general result in my opinion. This point is discussed by many studies. I would rephrase or delete this point.*

   This has been added in lines 420-421 of the revised manuscript as :
   This result is consistent and reaffirms findings from previous studies (e.g, Liao et al., 2021 ; Gallego et al., 2020 ; Hauck and Völker, 2015) ;

**References**

Liao, Enhui, Resplandy, Laure, Liu, Junjie & Bowman, Kevin W. (**2020**). "Amplification of the Ocean Carbon Sink During El Niños : Role of Poleward Ekman Transport and Influence on Atmospheric CO2". Global Biogeochemical Cycles. Vol. 34. no. 9, e2020GB006574.

Liao, Enhui, Resplandy, Laure, Liu, Junjie & Bowman, Kevin W. (**2021**). "Future Weakening of the ENSO Ocean Carbon Buffer Under Anthropogenic Forcing". Geophysical Research Letters. Vol. 48. no. 18, e2021GL094021.

Takahashi, Taro, Olafsson, Jon, Goddard, John G., Chipman, David W. & Sutherland, S. C. (**1993**). "Seasonal variation of $CO_2$ and nutrients in the high-latitude surface oceans : A comparative study". Global Biogeochemical Cycles. Vol. 7. no. 4, p. 843-878. eprint : https://agupubs.onlinelibrary.wiley.com/doi/pdf/10.1029/93GB02263.

Wanninkhof, Rik (**2014**). "Relationship between wind speed and gas exchange over the ocean revisited". Limnology and Oceanography : Methods. Vol. 12. no. 6, p. 351-362.

"Contrasting projection of the ENSO-driven $CO_2$ flux variability in the Equatorial Pacific under high warming scenario"
by P. Vaittinada Ayar et al.

We first would like to thank the anonymous reviewer for her/his thorough reading and very positive and constructive comments. We tried to take them into account as much as possible. A detailed point-by-point reply to these comments is provided below. Changes in the manuscript are indicated in blue.

**Answer to Referee #2 :**

**General comments :**

1) *Figure 5 and Line 206 : "This reversal is thus independent on the performance of the model over the contemporary period, though the models in the first row tend to simulate lower than observed CO2 flux anomaly variability."*

*Firstly, after reading the manuscript, the results suggested that the reversal behavior was indeed induced by the model performance in the contemporary period. Authors should modify or clarify this sentence.*

**Authors' response :** Thank you for pointing this out. The sentence was meant to state the ability of a model to reproduce the oberserved level of correlation over the contemporary period does not give any indication about the reversal behaviour.

This sentence has been modified in lines 211-212 of the revised manuscript as :
This reversal is thus independent from the model ability to reproduce the observed correlation over the contemporary period, though the models in the first row tend to simulate lower than observed $CO_2$ flux anomaly variability.

*Secondly, when looking at figure 5, the lower CO2 flux variability in the "reversed" ESMs than in the "preserved" ESMs is a striking feature. I would like to see some discussion about the influence (or the relationship) of this feature with the conclusions. For example, could the historical low CO2 flux variability in the "reversed" ESMs be related to their higher carbon uptake than in the "preserved" ESMs ? Authors focused on the understanding of the correlation between the annual CO2 flux and the ENSO index, but could some of their findings explain the variability in the amplitude of the simulated CO2 flux anomalies ? As a reminder, most models underestimated the CO2 flux variability (line 197 and Table 3) and according to the figure 5 this is more visible in the "reversed" ESMs.*

**Authors' response :** Thank you for this interesting point. We computed $k$ the gas transfer velocity multiplied by $K_0$ the solubility used to estimate $CO_2$ fluxes as F = $k * K_0$ *(pCO$_{2o}$ - pCO$_{2a}$). Figure R1 represents the $k * K_0$ and surface anomalies for each period, group of models and ENSO

phase. All the three preserved and reserved groups are able to reproduce the observed weakening in trade winds during El Niño, and vice versa during La Niña. The amplitude between ENSO phases is larger for the preserved models than the reversed ones. This can explain the higher amplitude variation between ENSO phase for the preserved models than the reverse ones (see Table 3 and Fig 5 of the article). Furthermore, we have added supplementary figure S6, depicting water column alkalinity concentration in the models, as compared to the observations. The three reversed ESMs simulate considerably high bias in alkalinity, consistent with high bias in carbonate ion (Fig. 9). The high alkalinity in the reversed models also contribute to the low contemporary $CO_2$ flux variation as it dampens the DIC-induced $pCO_2$ variability during different ENSO phases.

Figure R1 has been added to the supplementary material and additional discussion has been added in lines 243-254 of the revised manuscript as :

[Figure]

FIGURE R1 – El Niño and La Niña surface wind (*top* in m s$^{-1}$) and $k * K_0$ (*bottom* in mol C m$^{-2}$ yr$^{-2}$ atm$^{-1}$) mean anomalies for the reversed (*left*) and preserved (*right*) ESMs over the early historical (1851-1880), contemporary (1985-2014) and future (2071-2100) periods in the EP domain. Vertical bars represents ± one s.d. of the anomalies for the respective periods, groups of models and ENSO phases.

In addition to surface ocean $pCO_2$, $CO_2$ flux is estimated using atmospheric $pCO_2$ and wind solubility coefficient $k * K_0$ as :

$$\text{fgco2} = k * K_0 * (pCO_{2o} - pCO_{2a}) \tag{4}$$

$k$ represents the gas transfer velocity and $K_0$ the solubility coefficient [cf. Wanninkhof, 2014 ]. The anomalies of surface wind and product of $k * K_0$ for each period, group of models and ENSO phase are depicted in Fig. S8 of the supplementary material. The amplitude of both anomalies between ENSO phases is larger for the preserved models than the reversed ones, which partly explains the higher amplitude of $CO_2$ flux variability variation between ENSO phase for the preserved models than the reverse ones (see Table 3 and Fig. 5). However, for the respective groups the amplitudes between ENSO phases are not changing between given the analysed periods. This means that the wind variability can only have a marginal contribution to $CO_2$ fluxes variability and can not explain the behaviour of the reversed group models. In addition, we also note that the relatively low contemporary $CO_2$ flux variation in the reversed models is also partly attributed to the simulated high alkalinity bias in these models (see Supplemental Fig. S9), as high background alkalinity would dampen the DIC-induced $pCO_2$ variability during the different ENSO phases.

**Authors' response2 :** As stated in the article, the reversed ESMs simulate higher surface DIC increase (see Fig. 10) explaining that the reversed ESMs simulate more carbon uptakes than the preserved models over the transient simulation period. This is attributed to the higher surface and subsurface alkalinity and $CO_3^{2-}$ (see new Fig. S10 & S11 for ALK and Fig. 9&10 for $CO_3^{2-}$) concentration simulated by the reversed ESMs at the beginning of the transient simulation from surface to 300m depth. The considerably higher alkalinity (and carbonate ion) concentration in the reversed models yield watermass with higher buffer capacity, which allow them to uptake more atmospheric carbon in the future. This is the first order explanation for the projected higher surface $CO_3^{2-}$ reduction (see bottom panels of Fig. 9 and middle panel of Fig. 10). his higher buffer capacity also dampens the DIC-induced $pCO_2$ variability during ENSO phases which partly explains the smaller magnitude of $CO_2$ flux variability in the reversed models that was previously mentioned.

This has been addressed in lines 323-332 of the revised manuscript as :
The higher surface DIC increase is also illustrated in the right panel Fig. 10, depicting that the reversed ESMs simulate more carbon uptakes (or less cumulated DIC loss because the tropical Pacific is a mean outgassing system) than the preserved models over the transient simulation period. This is attributed to the higher surface and subsurface alkalinity and $CO_3^{2-}$ (see Figs. S10 and S11 for ALK and bottom panels of Fig. 9 and left panel of Fig. 10 for $CO_3^{2-}$) concentration simulated by the reversed ESMs at the beginning of the transient simulation from surface to 300m depth. The considerably higher alkalinity (and carbonate ion) concentration in the reversed models yield watermass with higher buffer capacity, which allow them to uptake more atmospheric carbon in the future. This is the first order explanation for the projected higher surface $CO_3^{2-}$ reduction (see bottom panels of Fig. 9 and middle panel of  Fig. 10). This higher buffer capacity also dampens the DIC-induced $pCO_2$ variability during ENSO phases which partly explains the smaller magnitude of $CO_2$ flux variability in the reversed models that was previously mentioned.

2) *Authors estimated the depth of the thermocline (line 105) but their discussion and conclusions focused on the stratification (or the vertical gradient), which are two different concepts. Although there is no difference between the two ESM groups in term of "thermocline depth" (line 306) the*

*vertical stratification might be different. Therefore, could authors replace their "thermocline depth" estimate with a stratification estimate.*

**Authors' response :** Following the reviewer good suggestion, we have calculated in situ density ($\rho$) from each ESM's potential temperature and practical salinity (after conversion to absolute salinity and conservative temperature) following TEOS-10 standards [FEISTEL, 2008] and using R "gsw" (https ://cran.r-project.org/web/packages/ gsw/index.html). Three-dimensional $\rho$ fields have been area-weighted over the EP region. We use a Stratification Index (SI) based on SGUBIN et al., 2017 to characterise the stratification of the water column from surface to 500m :

$$SI = \sum_{i=1}^{25} \rho^{Z_i} - \rho^{Z_0}$$

where $Z_0$ is the sea surface and $Z_i = Z_{i-1} + 20$ and $i \in [1, \cdots, 25]$.

As shown in Figure R2 below, SI does not provide distinguishable pattern between both groups of models. The estimated SI yield similar information  as thermocline depth concerning the upwelling, namely an increasing stratification in the future indicating a decrease in the upwelling. Higher stratification during El Niño events (indicating weaker upwelling state) and vice versa during La Niña, is maintained in the future.

Figure R2 and SI definition and formulation have been added to the supplementary material and thermocline time series have been removed from it. Modifications have been made in lines 344-351 of the revised manuscript as :

ENSO-induced upwelling variability alters the surface DIC anomalies.  Figure S13 of the supplemental material depicts time-series of the average Stratification Index (SI) computed over the EP domain (see supplemental for the definition and formulation). There is no significant difference in the  SI evolution between the reversed and preserved ESM groups. The  SI is expected to increase toward the end of the 21st century, consistent with future warmer upper layer and  weaker upwelling. In all ESMs, the  stratification variation due to ENSO, *i.e.*  higher stratification during El Niño events (indicating  weaker upwelling state) and vice versa during La Niña, is maintained in the future.  Despite increasing future stratification and shallowing of thermocline depth (see Fig. 9), the ENSO-driven surface DIC variation in all ESMs (anomalously lower DIC during El Niño and higher DIC during  La Niña) is also maintained in the future (see Fig.  S14).

[Figure]

FIGURE R2 – Time-series of average SI (in kg m$^{-3}$) from 1985 to 2100. The blue and red colours indicates the occurrence of the La Niña and El Niño regimes. The decadal trend is given for each model. Models names are given in green for the models with shifting correlation sign, in orange for those maintaining the negative correlation and black the others. The SI standard deviation ($\sigma$) over the early historical (1851-1880) and future (2071-2100 ) periods are given for each model.

**Minor comments :**

3) *Line 28 : "...the Equatorial Pacific CO2 flux represents the dominant mode of variability of the global oceanic CO2 flux variations (Wetzel et al., 2005 ; Resplandy et al., 2015...". According to Resplandy et al. (2015), for some ESMs, the Southern Ocean can also be the dominant mode of variability of the global oceanic CO2 flux variations.*

   **Authors' response :** It has been added in the revised manuscript in lines 29-30.
   Some ESMs also show the $CO_2$ flux in the Southern Ocean as the dominant mode (Resplandy et al., 2015).

4) *Line 110 – At which temporal resolution is the thermocline depth estimated ? Monthly ?*

   **Authors' response :** It has been estimated at monthly resolution. We have included this information in the revised manuscript in lines 114.

5) *Line 175 : "Note that the observed average is the result of the climatology over the 2004-2017 period while the average for CMIP6 is computed over 30 years (1985-2014)." Could authors calculate the CMIP6 climatology using the same period (i.e., 2004-2017) ? If not, this information should be included in Figure 3.*

   **Authors' response :** We prefer to keep the predefined 1985-2014 contemporary period to ensure more robust estimate of mean state (longer time scale) and to be consistent with the remaining analysis done throughout the paper. The fact that the observed average is the result of the climatology over the 2004-2017 period is reminded Figure 3 caption.

6) *Line 188 : "The correlation between annual CO2 flux anomaly and annual ENSO index is given for the models for each 30-year sliding window over the 1850-2100 period." Why did author choose a 30-year sliding window ? Is it the observational period ? This information needs to be added.*

   **Authors' response :** $CO_2$ flux observation is available over the 1982-2015 period ; 1985-2014 is a 30-years period chosen within that period. 30-years window is the typical climatological window used in numerous studies. This has been clarified in the revised manuscript in lines 193.
   (30-year is a typical the climatological window used in numerous studies)

FEISTEL, Rainer (**2008**). "A Gibbs function for seawater thermodynamics for -6 to 80°C and salinity up to 120g kg$^{-1}$". Deep Sea Research Part I : Oceanographic Research Papers.  Vol. 55. no. 12, p. 1639-1671.

SGUBIN, Giovanni, SWINGEDOUW, Didier, DRIJFHOUT, Sybren, MARY, Yannick & BENNABI, Amine (**2017**). "Abrupt cooling over the North Atlantic in modern climate models".  Nature Communications.  Vol. 8. no. 14375.

WANNINKHOF, Rik (**2014**). "Relationship between wind speed and gas exchange over the ocean revisited".  Limnology and Oceanography : Methods.  Vol. 12. no. 6, p. 351-362.

"Contrasting projection of the ENSO-driven $CO_2$ flux variability in the Equatorial Pacific under high warming scenario"
by P. Vaittinada Ayar et al.

We first would like to thank the anonymous reviewer for her/his thorough reading and very positive and constructive comments. We tried to take them into account as much as possible. A detailed point-by-point reply to these comments is provided below. Changes in the manuscript are indicated in blue.

**Answer to Referee #3 :**

**Major comments :**

- *Even though the authors are looking at the response of coupled models, the authors have ignored any changes or even providing statements about the atmospheric response (except Lines 340-344). There is no mention of changes in trade winds and/or changes in conditions of the air-sea interface due to the weakening of the easterly trade winds (during El Niño, for example). To me, this is a key ingredient that is missing from the study. This is a CMIP-based study and since the tropical ocean-atmosphere are strongly coupled with each other, the authors do need to provide qualitative statements about how atmospheric conditions across the ESMs (preserved vs. reserved) evolve that impact the oceanic ENSO response. Quantitative analyses regarding changes in atmospheric winds across the study time periods (or a figure or two) would be better, but I recognize that a quantitative evaluation of dynamical wind response is not a trivial task.*

  **Authors' response :** Thank you for this comment. In response to reviewer #1 request, we have calculated the simulated wind-solubility coefficient ($k * K_0$) during different ENSO phases and how they evolve in the projections for the two model groups. This information, together with the respective surface wind anomalies, are now presented in Fig. R1. It shows that both model groups are able to simulate the weakening of easterly trade winds during El Niño, and vice versa during La Nina under the contemporary period. In addition, the anomaly amplitudes are generally stronger in the preserved models, which is consistent with the higher amplitude of $CO_2$ fluxes variability in the preserved models (see Table 3 and Fig. 5). However, for the respective groups the amplitude of the surface wind anomalies between ENSO phases is not changing between given the periods, suggesting that the wind variability can only marginally contribute to $CO_2$ fluxes variability and can not explain the behaviour of the reversed group models.

  Figure R1 has been added to the supplementary material and this question has been addressed in lines 243-251 of the revised manuscript as :

  In addition to surface ocean $pCO_2$, $CO_2$ flux is estimated using atmospheric $pCO_2$ and wind solubility coefficient $k * K_0$ as :

  $$\text{fgco2} = k * K_0 * (p\text{CO}_{2o} - p\text{CO}_{2a}) \tag{4}$$

[Figure]

FIGURE R1 – El Niño and La Niña surface wind (*top* in m s$^{-1}$) and $k * K_0$ (*bottom* in mol C m$^{-2}$ yr$^{-2}$ atm$^{-1}$) mean anomalies for the reversed (*left*) and preserved (*right*) ESMs over the early historical (1851-1880), contemporary (1985-2014) and future (2071-2100) periods in the EP domain. Vertical bars represents $\pm$ one s.d. of the anomalies for the respective periods, groups of models and ENSO phases.

$k$ represents the gas transfer velocity and $K_0$ the solubility coefficient [cf. WANNINKHOF, 2014 ]. The anomalies of surface wind and product of $k * K_0$ for each period, group of models and ENSO phase are depicted in Fig. S8 of the supplementary material. The amplitude of both anomalies between ENSO phases is larger for the preserved models than the reversed ones, which partly explains the higher amplitude of $CO_2$ flux variability variation between ENSO phase for the preserved models than the reverse ones (see Table 3 and Fig. 5). However, for the respective groups the amplitudes between ENSO phases are not changing between given the analysed periods. This means that the wind variability can only have a marginal contribution to $CO_2$ fluxes variability and can not explain the behaviour of the reversed group models.

- *The authors should consider evaluation of the models for specific ENSO cases - strong El Niño or strong La Niña years. Figure 5 provides a first indication that the "preserved" ESMs agree better with the observations than the reversed ESMs. But the comparison is noisy, and it may be better to examine specific strong and very strong ENSO events between 1950-2014. Approximately 10 such events can be identified for both El Niño and La Niña conditions that should allow robust statements on which of the two groups of ESMs (preserved vs reserved) validate better against observations.*

**Authors' response :** Thank you for this comment. Models have been selected according to the correlation between annual $CO_2$ flux anomalies and Niño 3.4 index. We agree if the comparison is solely based on Figure 5, it is noisy. However the aim of our study is to identify common responses or pattern or changes in ENSO-CO2 flux variability among CMIP6 models. To keep it simple, we have applied one s.d. of Nino34 index to classify El Nino (Nino34>1s.d.) or La Nina (Nino34<1s.d.) months. As suggested, we have now explored whether or not we can get clearer comparison when taking into account only extreme ENSO events, (i.e. using 1.5 s.d. criteria). Figures RX and RY show contemporary SST and CO2 flux anomalies from observations and models using 1.5 s.d. criteria for distinguishing El Nino vs La Nina events. Except for one model (CNRM-ESM2-1), we do not see any significant difference in the spatial patterns when compared to Figs. S1 and S2 in the manuscript.

[Figure]

FIGURE R2 – CMIP6 ensemble SST (in °C) average anomalies over the 1985-2014 contemporary period for the La Niña, El Niño and the moderate regimes.

[Figure]

FIGURE R3 – CMIP6 ensemble sea-air $CO_2$ fluxes (in mol C m$^{-2}$ yr$^{-1}$) average anomalies over the 1985-2014 contemporary period for the La Niña, El Niño and the moderate regimes.

**Minor comments :**

- *Line 2 – change to 'over the tropical ocean less carbon is released during El Niño...'*

  **Authors' response :** Done

- *Line 56 – it is hard to interpret what the authors mean by the phrase 'an end member future projection'. While it becomes clear eventually that the authors are referring to the high-emission scenario, maybe add a sentence or two here to clarify this phrase for the benefit of the reader.*

  **Authors' response :** Modified in line 59 of the revised manuscript as :

  in future projection run under a high warming scenarios

- *Line 97 – please check the grammar and punctuation*

  **Authors' response :** Done

- *Lines 145-146 and Lines 337-339 - it is a bit strange that while the authors define a classical Niño 3.4 domain (Lines 99-100), the study area is subsequently shifted to different longitudes. This matters because not all El Niños are similar and whether we are looking at an EP or a CP El Niño should have implications for the findings of this study. Did the authors consider evaluating the model simulations based on different El Niño types?*

  **Authors' response :** Niño 3.4 domain is only used to compute Niño 3.4 index. This index is used to categorised month into El Niño or La Niña regime see section 2.2. The EP domain 2°S-2°N and 180°-260°E is identified as the common domain where the models and observation show the largest change in SST between ENSO phases see section 2.5. In this study, the aim is to examine the long-term average of ENSO-related CO2 flux variability and how it response in the high $CO_2$ future. As such, we think discriminating into more specific El Nino events (e.g., EP vs CP El Nino) is beyond the scope of this paper.

- *I would strongly encourage a modified version of Figure 11 – again, instead of looking at 1850-2100, maybe pick a period or specific strong & very strong ENSO years, for which the authors can plot a 'best estimate' of air-sea CO 2 flux from observations and/or models (for example, see Ishii et al., 2014, Biogeosciences, https ://doi.org/10.5194/bg-11-709-2014). It would be interesting to see which of the ESMs actually fall in the region where surface $CO_3^{2-}$ concentration obs. and the most optimal estimate of air-sea CO2 fluxes overlap. Can we identify a subset within the 16 CMIP6 ESMs that validate better against the observations? This study has already laid the foundation for providing this key message, thereby really helping the improvement of future ESMs and CMIP simulations.*

  **Authors' response :** Thank you for this comment. We would like to clarify that Figure 11 is meant to illustrate if historical carbonate concentration can provide a constrain on the projected cumulated $CO_2$ fluxes over the historical and future period. In that case choosing specific strong or very strong ENSO year would not be helpful although it would provide similar insight that models, which overestimate the surface carbonate ion concentration tend to simulate high cumulative CO2 fluxes (see Figure R4 below). Figure R4 which is similar to Figure 11 but we added on the left panel the cumulated $CO_2$ flux over the observational period. It provides us the same relationship between carbonate and cumulated $CO_2$ fluxes over the 1850-2100 period. Over contemporary period, the majority of the models (10/14) are underestimating the observed cumulated sea-air $CO_2$ fluxes but reversed models simulate the largest underestimation.

[Figure]

FIGURE R4 – Average contemporary surface $CO_3^{2-}$ concentration (in µmol C $L^{-1}$) plotted against the cumulated sea-air $CO_2$ fluxes (in Pg C) in the EP region from 1985 to 2014 (*left*) and 1850 to 2100 (*right*). ρ is the correlation and reversed ESMs are marked in yellow and preserved ones in blue. The observations are given in brown lines with dashed lines being the carbonate observation error.

Figure R4 is the new Fig. 11 of the article and additional discussion has been added in lines 334-339 of the revised manuscript as :

Figure 11 shows contemporary surface carbonate concentration against the cumulated sea-air $CO_2$ flux from 1850 to 2100 over the 1985-2014 and 1850-2100 periods over EP for all the models except the MPI models. The correlation at Correlation at 0.65 and 0.67 indicates indicate that the carbonate concentration is a good indicator of the buffering capacity of the model : the higher the carbonate the lower the cumulated $CO_2$ outgassing (ie. more carbon uptakes). The preserved ESMs are less biased in terms of carbonate concentration and cumulated $CO_2$ flux over the contemporary period, which tend to indicate that their behaviour should be more reliable.

WANNINKHOF, Rik (**2014**). "Relationship between wind speed and gas exchange over the ocean revisited". Limnology and Oceanography : Methods. Vol. 12. no. 6, p. 351-362.

---

## Author Response (AR2)

"Contrasting projection of the ENSO-driven CO$_2$ flux variability in the Equatorial Pacific under high warming scenario" by P. Vaittinada Ayar et al.

We first would like to thank the Dr.Gabriele Messori his thorough reading and swift handling of this manuscript. We also thank you for considering this work for publication. We replied the editor comment and changes in the manuscript are indicated in blue.

**Answer to the Editor :**

*Dear Authors,*

*Following your revisions, the manuscript is suitable for publication in ESD. The only minor point I would ask you to implement is to add a sentence to the text addressing the analysis you have conducted in Figs. R2 and R3, which is a good test of the robustness of your results.*

*Best Regards,*
*Gabriele Messori*

**Authors' response :** Thank you for this point. This question has been addressed in lines 171-173 of the revised manuscript as :

Same results a are obtained when looking for more extreme El Niño and La Niña regimes (for months with Niño34 index respectively above 1.5 s.d. and below -1.5 s.d., not shown).

Thank you again for your consideration.

With Regards,

Pradeebane Vaittinada Ayar
On behalf of all the co-authors